# Gasdermin D pores are dynamically regulated by local phosphoinositide circuitry

Ana Beatriz Santa Cruz Garcia[1], Kevin P. Schnur[1,2], Asrar B. Malik [1] & Gary C. H. Mo [1,2 ✉]

Gasdermin D forms large, ~21 nm diameter pores in the plasma membrane to drive the cell death program pyroptosis. These pores are thought to be permanently open, and the resultant osmotic imbalance is thought to be highly damaging. Yet some cells mitigate and survive pore formation, suggesting an undiscovered layer of regulation over the function of these pores. However, no methods exist to directly reveal these mechanistic details. Here, we combine optogenetic tools, live cell fluorescence biosensing, and electrophysiology to demonstrate that gasdermin pores display phosphoinositide-dependent dynamics. We quantify repeated and fast opening-closing of these pores on the tens of seconds timescale, visualize the dynamic pore geometry, and identify the signaling that controls dynamic pore activity. The identification of this circuit allows pharmacological tuning of pyroptosis and control of inflammatory cytokine release by living cells.

[1] Department of Pharmacology and Regenerative Medicine, University of Illinois College of Medicine, Chicago, IL, USA. [2] Richard and Loan Hill Department of Biomedical Engineering, University of Illinois at Chicago, Chicago, IL, USA. ✉email: gmo@uic.edu

The cell death program pyroptosis plays an important role in the development and homeostasis of multicellular organisms[1,2]. It is also a root cause for inflammatory damage in tissues ranging from the endothelium[3] to cardio-vascular tissues[4]. Canonical upstream activation of caspase-1 in humans and caspase-11 in mice triggers this lytic program via the cleavage of gasdermins[5–8]. Activated Gasdermin D (GSDMD) forms ~21 nm diameter oligomeric pores in the plasma membrane[2,9–12], causing osmotic imbalance, cell lysis, and driving the release of pro-inflammatory cytokines IL-1β and IL-18[13] to promote downstream T cell and macrophage activation responses. Consistent with this role in cell death, gasdermin knockout[14] and inhibition through irreversible protein modification[15,16] prevented inflammation, and upregulation of pore formation in immune cells contributed to tumor suppression[17–19]. Yet unusual for a death program, pyroptosis is not always deadly. It is separable from cell death[20], can be overridden[21], and has been shown to collaborate with apoptotic signaling[22,23]. Large amounts of cytokines can be released from apparently intact and viable cells after gasdermin activation[24,25]. Furthermore, the apparent immediate danger posed by these large pores is contrasted by their slow rate of nonspecific removal from the membrane[26].

It is clear that these pores possess an outstanding property that dampens their lethality. Specifically, it has not been explored whether gasdermin pores are dynamic. We hypothesized that pyroptotic pores can be halted. In this concept, biochemical signals can shut these pores after their formation and resume by opening on the short seconds/minutes time scale, thereby modulating the progression of pyroptosis. If true, such a mechanism would allow the cells to simply regulate and escape pyroptosis after activation, which would, in turn, constitute a new route to reversibly control these pores and hence the downstream inflammatory response. Unlike stably integrated membrane ion channels, gasdermin pores require activation and assembly. However, there are currently no tools to study the dynamics of large oligomeric pores spanning their activation, formation, and function in live cells. In this work, through a combination of electrophysiology, optogenetic design, live-cell fluorescent biosensing, and image analysis, we demonstrate that GSDMD pores are indeed dynamic structures. Exploring their kinetics and intrinsic biochemical circuitry in model bilayers and live-cell plasma membrane, we show that pore-mediated calcium influx modulates pore opening/closing kinetics through phosphoinositide metabolism. By defining and perturbing the parameters of this mechanism, we reveal a pharmacological approach to up- or down-regulate GSDMD pore activity, and hence cellular cytokine release and pyroptotic cell death.

## Results

### Optogenetic activation of GSDMD forms pores and recapitulates the phenotype of activated macrophages and fibroblasts.

Direct and exogenous activation of GSDMD in live cells is required to study the putative dynamics of gasdermin pores. Thus, we developed an optogenetically activatable human GSDMD in which the C-terminal autoinhibitory domain is cleaved[27] and released upon blue light illumination (Fig. 1a, scheme), allowing precise and orthogonal analysis of GSDMD pore dynamics. As GSDMD pores mediate ion influx and efflux[26,28], we examined whether the hypothesized pore dynamics would be reflected by transient calcium responses. In transiently transfected RAW264.7 and mouse bone marrow-derived macrophages (BMDM), activation of the diffusible optogenetic GSDMD (termed PhoDer) initiated clearly localized transient calcium flares (Supplementary Fig. 1a and Supplementary Movies 1, 2). In HeLa cells expressing PhoDer, optogenetic activation similarly caused spontaneous calcium flares at sporadic locations on the plasma membrane, which were visible via membrane-targeted jRCaMP1b[29] (Fig. 1b). Upon simple tuning, overt activation was avoided, cells did not lyse for tens of minutes, and we observed calcium flare events, followed by a gradual increase in whole-cell calcium, and finally calcium efflux and cell lysis as reported for pyroptosis[20] (Fig. 1c and Supplementary Movie 3). Both flares and whole-cell calcium increases were absent when extracellular calcium was withheld (Fig. 1d), suggesting that intracellular calcium stores did not contribute to these events. All cells ($n = 81$ biologically independent cells in $n > 5$ independent experiments) showed localized calcium flares and widespread whole-cell calcium fluctuations (Fig. 1c); the large standard deviation indicated that the progression was asynchronous across the cell population (Fig. 1d). With further tuning, rapid PhoDer activation reproduced the severe blebbing "bubble" formation observed after nigericin-induced NLRP3 inflammasome activation in primed BMDMs (Supplementary Fig. 1b). Controls lacking PhoDer expression lacked such calcium response (Supplementary Fig. 1c). The design of PhoDer involved no other mechanism except the simple release of monomeric GSDMD autoinhibition, suggesting that GSDMD alone was responsible for the calcium flares in macrophages and fibroblasts.

To further address whether the calcium dynamics primarily reflected that of GSDMD pores, we directly recorded their actions via whole-cell patching in living endothelial cells, which also undergo pyroptosis[3]. Only the injection of activated, recombinant GSDMD proteins into these cells showed single pore currents that again marked repeated fast pore openings, stable current, and quick pore closures, whereas controls showed little response ($n = 15$ for each, Supplementary Fig. 2a, b). Additionally, as befitting a large pore, the GSDMD pore opening/closing dynamics were not restricted to calcium observations. We visualized calcium flares occurring in coincidence with the transient flux of a large fluorogenic molecule, within the same spatiotemporal context in living cells (Supplementary Fig. 3 and Supplementary Information). Furthermore, expression of membrane-targeted GSDMD-C (C-terminal domain of GSDMD) significantly dampened the optogenetically-induced calcium response (Supplementary Fig. 4), suggesting that post-activation, GSDMD did not bind other partners and remained accessible to cognate inhibition. Therefore, observations across multiple platforms and perspectives show that GSDMD pores are highly dynamic in the live-cell plasma membranes.

### Gasdermin D pores close intact without membrane disruption.

GSDMD oligomers form large pores without obstructions, unlike ion channels[6]. The above results demonstrated that pore kinetics and size are not conducive to live-cell super-resolution imaging, making it difficult to address the closure of such supra-structures. However, the calcium influx does create an expanding wavefront while the pore remained open. We thus surmised that the perimeter of this wavefront could be resolved to approximate the underlying, dynamic pore geometry. Utilizing PhoDer in combination with an algorithm similar to photochromic Stochastic Optical Fluctuation Imaging (pcSOFI[30,31], Supplementary Information), we highlighted the location of high calcium-change to visualize each single pore during its open/close actions. The images were sufficiently sensitive to discern the geometry of pores above the slow-evolving intracellular calcium signals. We routinely generated images that, while diffraction-limited, emphasized flares at a signal-to-background ratio exceeding three orders of magnitude. This enabled us to algorithmically detect and quantify GSDMD pore dynamics.

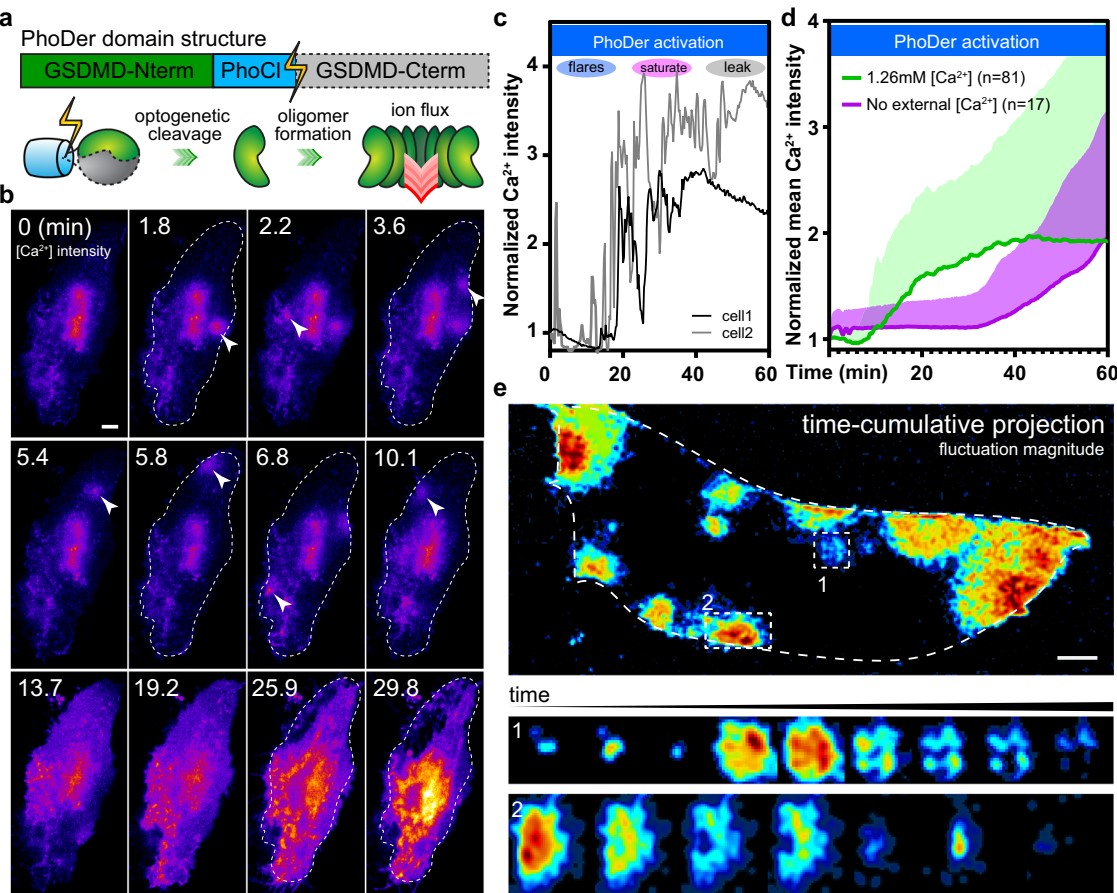

**Fig. 1 Optogenetic GSDMD pore activation in live cells shows transient local calcium flares. a** The domain structure and scheme for the optogenetically activatable GSDMD, PhoDer. C-terminal autoinhibition is cleaved after blue light exposure, liberating the N-terminal pore domain for membrane insertion. **b** Representative calcium biosensing images showing the activation of PhoDer in HeLa cells ($n = 81$) induced repeated calcium flares (arrowheads); dashed outlines in each panel indicate the outline of the single cell. **c** Intracellular calcium progression in representative cells over 1 h optogenetic activation time course could be categorized into the flare, saturation, and leakage phases. **d** Whole-cell average calcium showed the PhoDer-induced accumulation of intracellular calcium; line: mean value, shaded region: standard deviation. Control cells where PhoDer was activated without extracellular calcium did not show such an increase. **e** Fluctuation analysis yielded high contrast images of flares; dashed curved outline shows the extent of the single cell. Top: time-projection shows the history of multiple flares within the same cell. Two are highlighted by dashed squares. Middle/bottom: representative evolution of two flares, where the pore opens, shuts, reopens, and finally closes again. The calcium current in each time point centers on approximately the same cellular locale. Scale bar: 10 μm.

First, by accumulating the flare events in a representative cell over time into a single image, we showed that flares did not have a simple spatial preference but occurred across the plasma membrane (Fig. 1e, top panel). We next analyzed the kinetics as well as the spatial extent of the calcium flares using a verified subset. On average, calcium flares maximized in 0.35 min, and pores were open for an average of 1.55 min, reaching a mean diameter of ~10 μm ($n = 21$ pores). The pore closing was well-approximated by a kinetic inhibitory response from cumulative calcium; we estimated that pore closing displayed a cooperative coefficient of 0.73 (mean of $n = 21$; $R^2 = 0.996$). Interestingly, the same pore could re-open in-place in the live-cell membrane (Fig. 1e, lower panels), and ~57% of the pores persisted after the mean open time and reopened. Faster calcium flares with an average $t_{1/2}$ of 3.5 s could be found using higher frequency imaging (Supplementary Fig. 5). Image analysis showed that optogenetically-activated calcium signals emanated largely from circular pores (Fig. 1e). This evolved from a single spark in the center to a ring of high flux with the propagation of the calcium flare. We saw occluded, semi-circular flares when a pore opened near the edge of the cell (Fig. 1e, lower panels). Otherwise, we did not observe asymmetrical events or catastrophic membrane leakage, suggesting pores did not close via disassembly. Within these geometric limits, the average

circularity (minor/major radii ratio) of the flares was 0.75. Thus, in live-cell membranes, GSDMD pores likely opened and closed neatly as ellipsoidal, intact oligomer pores.

**GSDMD pore dynamics is phosphoinositide-dependent**. We surmised that the dynamics of these oligomeric pores may stem from the thermodynamic influences of the lipid bilayer environment. To study the role of the bilayer, we next employed in vitro electrophysiology (Fig. 2a, scheme) to examine the kinetics of reconstituted GSDMD pore in a free-standing membrane bilayer. Following the formation of a stable phospholipid bilayer (POPE/POPC; Supplementary Information), all controls (bilayer-only, addition of either recombinant GSDMD or caspase-1 alone) showed no change in micro-current recordings ($n = 20$ bilayers each, Fig. 2b). However, upon the addition of activated GSDMD (mixture with caspase-1 to cleave the C-terminal autoinhibition domain), we detected minute current fluctuations suggesting membrane-protein interactions that preceded pore formation ($n = 20$ bilayers, Fig. 2b, highlighted region). On average, these micro-currents occurred $27.2 \pm 2.7$ min after protein introduction. Shortly thereafter, we saw a single pore ion flow characterized by macro-currents. Strikingly, the macro-current traces also revealed

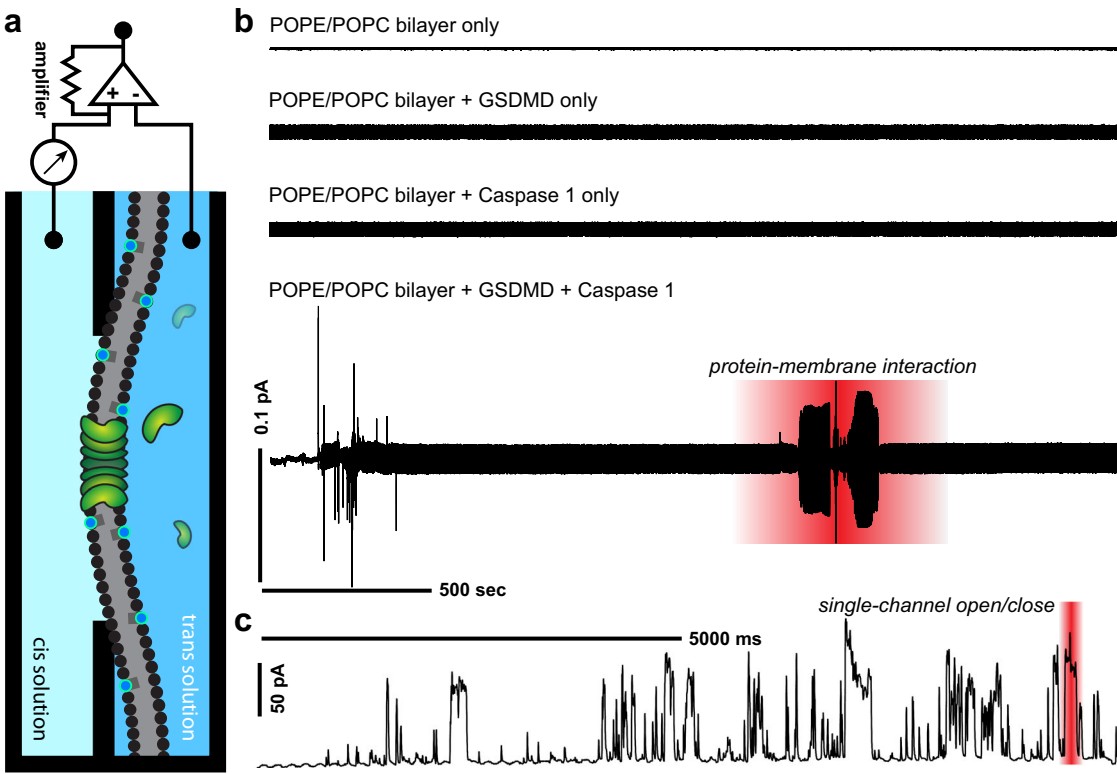

**Fig. 2 In vitro characterization of GSDMD pores in lipid bilayers reveals repeated spontaneous pore opening and closing events. a** Schematic for electrophysiological setup used to monitor reconstituted GSDMD pores in a planar lipid bilayer in vitro. Silver-silver chloride electrodes connecting *cis* and *trans* compartments conducted the current, which was subsequently amplified and recorded. **b** Representative traces of GSDMD protein–membrane interaction and single pore formation events in lipid bilayers using a gap-free protocol. From the top of the panel: recordings in controls including baseline bilayer (POPE/POPC) alone, baseline bilayer with the addition of GSDMD protein only, and baseline bilayer with the addition of caspase-1 only, were absent of current activity. Bottom of panel: addition of activated GSDMD to the baseline bilayer demonstrated protein–membrane interaction characterized by fluctuating micro-currents. **c** Representative trace of single pore activity showing spontaneous and repeated opening and closing events across the GSDMD pore.

that the recombinant pores closed and opened repeatedly in the model bilayers as we observed in live-cell membranes (Fig. 2c). Further analyses clearly contrasted the single pore events from those of unitary, single ion channels. The peak currents of single GSDMD pores appeared more varied compared to binary ON/OFF states typical of ion channels. We attribute the nonuniform peak currents to an "analog" character of these pores, whose oligomeric nature means that both the maximum pore size and the pore supra-structure are dynamic. The histogram of peak currents showed a broad distribution with a mean current of 32 pA. A largely uniform dwell time distribution indicated little preference in how long they remained open within the time window; an open probability of ~0.2 indicated a slight preference for the closed state. A ramp protocol showed dual rectification (Supplementary Fig. 6), indicating these pores were not inherently ion-selective. Thus, the active GSDMD pores in artificial membranes spontaneously opened and closed, consistent with the above observation of calcium flares in live cells.

The observed kinetics of the live-cell pores are compatible with local, calcium-dependent modification of the membrane lipid composition. This would in turn create the thermodynamic conditions necessary for GSDMD pore closure as a negative feedback response shortly after opening. Importantly, we observed the above dynamics in a membrane without phosphoinositide, which was presumed to be required for GSDMD pores[6]. This prompted us to examine whether the role of phosphoinositide is instead to control GSDMD dynamics. Phospholipase C (PLC) and phosphoinositide 3-kinase (PI3K) are two major

calcium-dependent phosphoinositide-modifying enzymes acting on the substrate PtdIns(4,5)P$_2$ (PIP2), present in the inner leaflet of the plasmalemma membrane[32]. Specific PLC activity requires calcium and is further activated via transient calcium entry;[33] PI3K is directly activated by calcium-calmodulin[34]. Under calcium stimulation, the enzymes PLC and PI3K promote the relative accumulation of diacylglycerol (DAG) and phosphatidylinositol triphosphate (PtdIns(3,4,5)P$_3$, or PIP3), respectively. Therefore, we first employed the reconstituted bilayer models to examine the consequences of shifting membrane PIP2 content to either PIP3 or DAG. The introduction of PIP2 (10%) to the POPE/POPC baseline bilayer composition decreased the time to observe protein–membrane interaction significantly to $2.4 \pm 0.5$ min (Fig. 3a); PIP2 inclusion also significantly reduced the mean current of these pores to 20 pA (compared to 32 pA in baseline bilayers). The presence of PIP2 made the pores highly dynamic, leading to a significant 57-fold increase in open/close events. The current histogram suggested that PIP2 addition reduced the diversity of pore conformations (Fig. 3d). However, PIP2 addition did not affect the open probability of these pores and the dwell time was similarly distributed compared to the baseline phospholipid bilayer (Fig. 3b, c). Overall, PIP2 inclusion directly enhanced the dynamics of GSDMD pores. Of note, the inner mitochondrion membrane lipid cardiolipin (CL)[8], another lipid target of GSDMD, also supported a pore dynamics profile similar to that of PIP2 (Supplementary Fig. 7).

We next examined the equilibrium condition where all PIP2 had been metabolized by PLC, substituting DAG for PIP2 in the

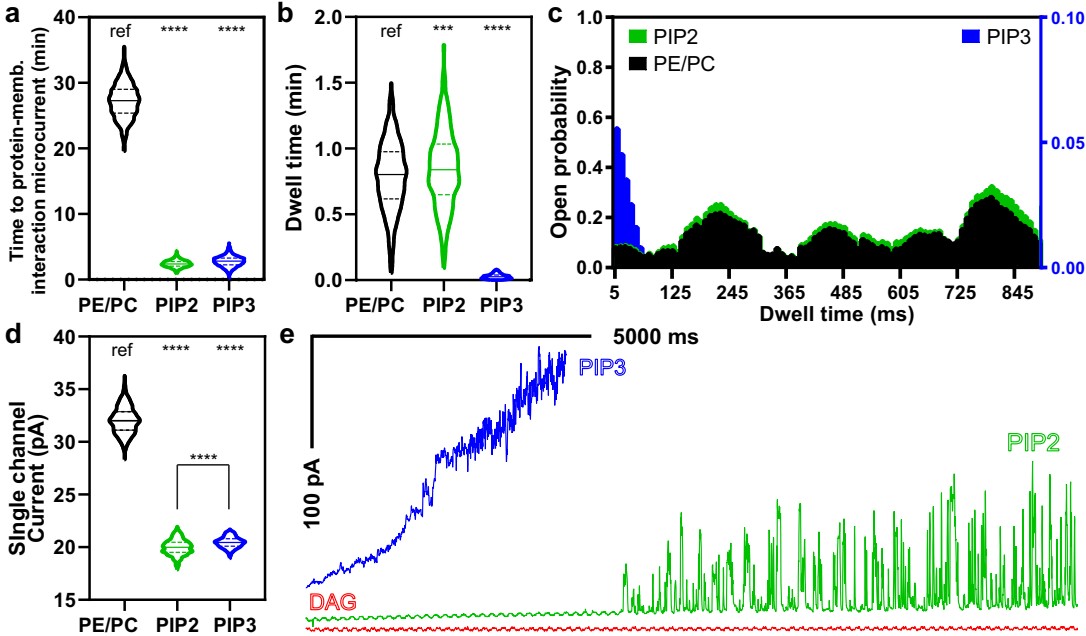

**Fig. 3 Phosphoinositide composition regulates GSDMD pore dynamics in model lipid bilayers. a** PI(4,5)P$_2$ (PIP2) and PI(3,4,5)P$_3$ (PIP3) content significantly shortened the time to observe GSDMD protein–membrane interaction compared to baseline phospholipid bilayer. **b** The dwell time (open duration) of GSDMD pores is similar between PE/PC and PIP2-containing bilayers; PIP3-containing bilayers induced apparently short dwell time but bilayer suffered strong osmotic pressure because pores could not close. **c** Open probability histogram of single pore events showing distinct responses to phosphoinositide composition. **d** Histogram of single pore open-close events in different bilayer compositions; either phosphoinositide significantly reduced the current. **e** Representative traces of single pore recording when PIP2 and its metabolites, PIP3 and diacylglycerol (DAG), were incorporated at the same relative concentrations. General color-shade in this figure: PIP2 (green), DAG (red), or PIP3 (blue). DAG incorporation did not show single pore activity despite detectible protein–membrane interaction, thus is absent from panels (**a**) to (**d**). For panels (**a**), (**b**), (**d**), the reported statistical significance was derived from one-way ANOVA with multiple comparisons; "ref" represents the reference; in all cases ****$p < 0.0001$; ns not significant. In (**b**), ***$p = 0.0002$.

lipid mixture while maintaining the fatty acid chain length (di-oleoyl) and relative concentration (10%). Despite observing micro-currents that signify protein–membrane interaction (Supplementary Fig. 8), we could not detect any single pore activity ($n = 15$ bilayers) and thus were unable to quantify any macroscopic current. When PIP2 was replaced with PIP3, GSDMD protein–membrane interaction remained as rapid as with PIP2 (Fig. 3a). The presence of PIP3, however, caused a rapid, cumulative increase in the macroscopic current that obscured clear, single pore open/close events ($n = 20$ bilayers, Fig. 3e). The few single events captured showed similar mean current (20.5 pA, Fig. 3d) but displayed a different conformational landscape and were significantly shorter-lived (Fig. 3c). Thus, PIP3 content kept GSDMD pores predominantly in the open state while in contrast, DAG content induced the closed state. We note that phosphoinositide content had little effect on the rectification behavior and hence pore selectivity (Supplementary Fig. 6, PIP2/PIP3).

To explore the structural basis of the phosphoinositide control, we next altered two putative phosphoinositide interaction sites and examined how they contributed to GSDMD pore dynamics. A simple alignment comparison between known gasdermin homologs[35,36] (Fig. 4a) allowed us to identify regions (I) R42/K43 and (II) K51/R53/K55 in hGSDMD as potential sites where membrane-inserted monomers may access the phosphoinositide headgroups; these sites were also occluded by the autoinhibitory C-terminal domain. Altering either charge or hydrogen-bonding potential, we made and expressed several PhoDer mutants in live HeLa cells, performed calcium biosensing, and automated flare quantifications as above. While these mutants were still capable of insertion and pore formation and thus showed calcium responses,

they displayed significantly reduced pore opening/closing dynamics (Fig. 4b, c). It also seemed possible to remove the saturating calcium response while still maintaining repeated open/close events (Fig. 4d). Charge-altering mutations at other putative lipid-binding sites could similarly change pore dynamics. Mutations at the well-conserved N-terminal region retained clear flares but showed much-reduced calcium magnitude (AE, Supplementary Fig. 9), while two mutants at another helical region diverged, either encumbering or abrogating the calcium response (BE1/BE2, Supplementary Fig. 9). Thus, lipid-binding sites from at least three regions on the GSDMD N-terminal domain contribute to pore dynamics and can be specifically utilized to tune these dynamics.

**Calcium-driven phosphoinositide metabolism regulates GSDMD pore dynamics and cytokine release.** We next perturbed the lipid enzymes PI3K/PLC and investigated whether they are actively used by living cells to tune and control GSDMD pore dynamics. Inhibition of PLC with U73122 (10 μM) prior to PhoDer activation (same optogenetic activation regime as in Fig. 1) caused a steep, cumulative rise in intracellular calcium in all cells ($n = 72$ biologically independent cells) where individual flares were rarely observable (Fig. 5a, b). The U73122 pretreated cells rapidly reached the saturation and lytic stage. In contrast, PhoDer activation in cells pretreated with the pan-PI3K inhibitor wortmannin (10 μM) showed a significant delay in calcium progression across many cells ($n = 39$ biologically independent cells, Fig. 5a, b). The delay of onset was further accompanied by significant changes in calcium flares dynamics. Analysis of these flares using the above fluctuation contrast imaging showed that pores under wortmannin pretreatment closed slowly and

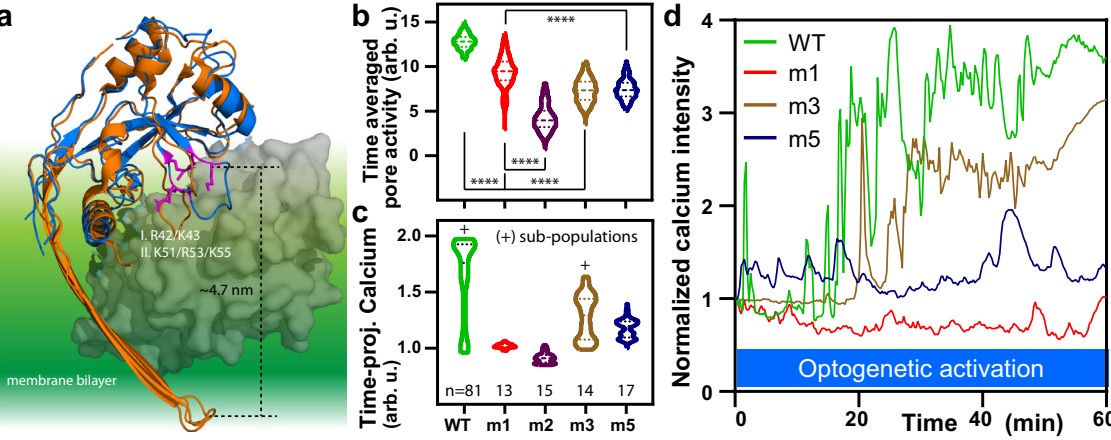

**Fig. 4 Mutants at two putative GSDMD phosphoinositide interaction sites alter the optogenetically-activated pore dynamics in living cells. a** Schematic for selection of putative phosphoinositide interaction site. mGSDMA3 structure (PDB: 6CB8, orange) was aligned with the partial structure of the N-terminal domain of hGSDMD (PDB: 6N9O, blue) and positively charged residues (magenta) are selected due to their location in two regions ~5 nm from the edge of the bilayer and burial by the autoinhibitory C-terminal (gray). Significant changes in **b** pore activity and **c** time-projected average calcium response were generated by altering the charge or hydrogen bond potential at the two indicated sites. Mutations were made in PhoDer at the corresponding GSDMD N-terminal sites. Mutant labels: m1: K51Q/K55Q; m2: K51E/K55E; m3: R42Q/K43Q; and m5: R42Q/K43Q/K51Q/K55Q. The + sign was used to denote two subpopulations that in time-projected terms indicated substantial calcium saturation. N numbers for each mutant is marked below the violin plots and applies to both (**b**) and (**c**). **d** Representative calcium response curve for mutants: m1 (overall inhibited calcium response); m3 (reduced flares but retained calcium saturation); and m5 (retained flares but reduced calcium saturation). Statistical significance was reported from two-way ANOVA with Tukey's multiple comparisons, ****$p < 0.0001$.

displayed a lower cooperative coefficient of 0.30 (mean of $n = 27$ pores; $R^2 = 0.998$). They reached a maximal calcium flux in 0.60 min, approximately twice that shown by pores on the untreated live-cell membrane. While the spatial extent these flares reached was not significantly different (12 μm, $n = 27$), flares in wortmannin-treated cells showed less temporal alignment and a longer effective average opening time of 3.30 min; subsequent reopening was also more delayed and spread out in time (Fig. 5c). We further utilized pixel-wise calcium fluctuations to compare the local pore activity across different treatments. U73122 pre-treated cells displayed the highest activity per locale compared to control, non-treated cells. Wortmannin pretreated cells showed a comparable mean activity per locale as non-pretreated cells, but with significantly less variability (Fig. 5d). Low, basal pixel-wise calcium activity was observed when extracellular calcium was withheld (ANOVA $p < 0.0001$; Fig. 5d). Together, these data showed that calcium-driven phosphoinositide composition shift significantly altered GSDMD pore dynamics.

Previous studies showed that gasdermins could be irreversibly inhibited[15,16]. Our data suggest that modulating GSDMD dynamics through phosphoinositide feedback reversibly controls pore function and in turn, the release of inflammatory cytokines. We thus examined the effects of regulating plasma membrane phosphoinositide dynamics on the release of pro-inflammatory IL-1β in LPS-stimulated mouse BMDM. Consistent with the above observations, IL-1β release 5 min after LPS exposure was significantly dampened by wortmannin and vice versa, enhanced by U73122 pretreatment (Fig. 6a). Following LPS-priming and nigericin-induced NLRP3 inflammasomal activation of GSDMD, we also observed a significant reduction of IL-1β release with PI3K inhibition (Fig. 6b). We further examined the effects of directly introducing phospholipids to alter live-cell inflammatory response. As predicted, dioleoylglycerol-rich liposomes treatment (Supplementary Information) significantly suppressed both the short-term LPS-induced (Fig. 6c) as well as the more prolonged nigericin-induced IL-1β release in BMDM (Fig. 6d). To probe the network integrity demanded by the GSDMD-calcium signaling, we further attempted to manipulate the PI3K/PLC balance

through activators. While the PLC activator m-3m3FBS[37] inhibited flare occurrence for a short time, its activation of other calcium signal[38] meant that, unlike PI3K inhibition, it did not limit LPS-stimulated IL-1β release (Supplementary Fig. 10). The peptidic PI3K activator 740 Y-P[39] also failed to elicit meaningful differences in calcium or cytokine release responses. These results demonstrate the highly specific and localized nature of the GSDMD-calcium circuit, further emphasizing the utility of the inhibitors described above. In aggregate, our data show that the calcium-influx driven phosphoinositide metabolism is a fundamental component of the GSDMD pore. PI3K/PLC enzymes form a major signaling circuit in regulating oligomeric Gasdermin D pore activity, through which we can modulate cytokine release and hence the strength of the downstream inflammatory signals (Fig. 6e).

## Discussion
Our studies show that GSDMD pores open and close, causing highly localized calcium flares distinct from other stimulated calcium oscillations seen in cells[40,41]. LPS is transported into the cells[3] through endocytosis (via CD14[42] and HMGB1[43]), where breaching into the cytosol[44] leads to NLRP3 inflammasome activation that in turn activates GSDMD[45]. Our optogenetic approach bypassed these complex steps altogether and allowed us to directly and quantitatively activate GSDMD. We showed that GSDMD dynamics influence cellular pyroptosis. While GSDMD-induced cell content efflux is separate from cell death[20], we clearly showed that cells lysed when the strength and duration of GSDMD activation was sustained. In contrast to assumptions, activated GSDMD did not require the phosphoinositide PIP2 to insert, form pores, and open and close. The present results including those from mutant GSDMD suggest that phosphoinositides alter the dynamic properties of the pores. Previous work showed that mature IL-1β collects in PIP2-rich plasma membrane regions[46]. Our findings support the concept that PIP2 enrichment marks the locations where GSDMD preferentially inserts and causes calcium flares. In vitro recordings and live-cell quantitative calcium biosensing suggest pore-mediated calcium entry engages

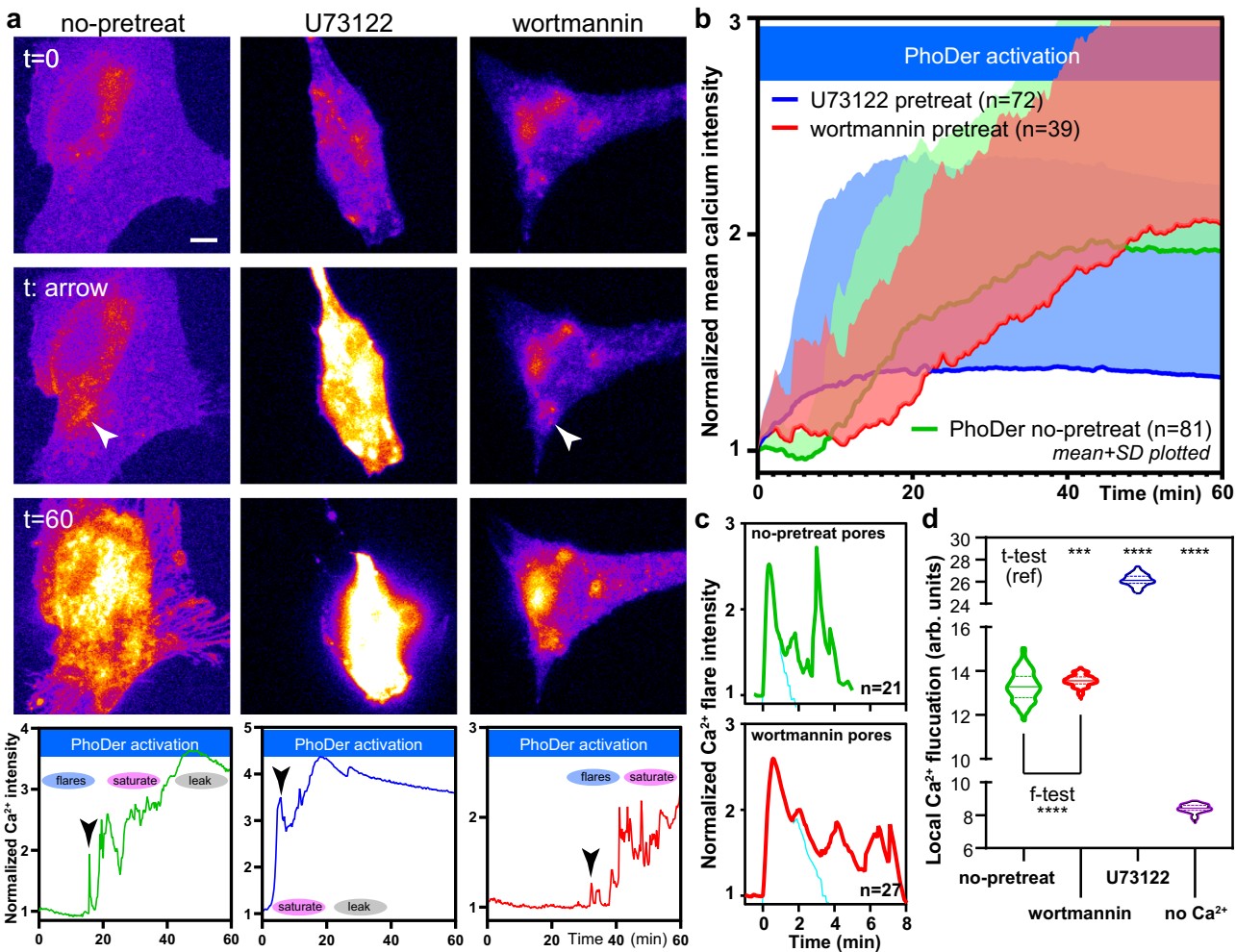

**Fig. 5 Inhibition of phosphoinositide pathway in live cells modulates GSDMD pore dynamics. a** Representative images and time course plots showing the effects of PLC and PI3K inhibitors on GSDMD pore functions. Left column: no pretreatment; middle column: 10 min 10 μM U73122 (PLC inhibitor); right column: 10 min 10 μM wortmannin (PI3K inhibitor). Scale bar: 10 μm. **b** Whole-cell average calcium levels over many cells support the results from single cell responses. The color scheme follows green: no pretreatment; blue: pretreated with U73122; and red: pretreated with wortmannin. Line: mean value; shaded: standard deviation (SD). **c** Changes in single calcium flare kinetics with or without pretreatment of wortmannin, which significantly slowed-down pore dynamics by a factor of 2. Cyan lines in each panel represent the flare signals subjected to the cumulative inhibitory model fit reported. **d** Pixel-wise calcium fluctuation show pore dynamics changes with phosphoinositide modulation. For panel (**d**), statistical significance was reported from individual paired *f*- and *t*-tests with two-tailed Welch's correction; "ref" represents the reference, ***$p = 0.0003$ (no-pretreat vs. wortmannin); ****$p < 0.0001$ (no-pretreat vs. U73122 and no-$Ca^{2+}$).

a feedback loop to change the phosphoinositide balance, effectively yielding self-closing GSDMD pores. While our optogenetic approach could tune and synchronize GSDMD activation, we did not observe synchronization of pore opening simultaneously across a cell population or indeed within the same single cell. Therefore, the local balance of calcium-dependent phosphoinositide-modifying enzymes appear to regulate GSDMD pore dynamics in a stochastic manner. In addition to the phosphoinositide action explored here, other downstream elements including second messengers and membrane potential may also contribute to altered pore confirmation and closure. Factors that influence intracellular calcium homeostasis, including extracellular calcium concentration, may also indirectly affect pore dynamics. Our approaches will guide future work to characterize the behavior and tunability of GSDMD pores in primary immune and other cell types.

Live cell calcium biosensing in conjunction with the optogenetic tool PhoDer proved highly sensitive for studying GSDMD activation. We could induce and detect the activation of a few monomers in a single cell, which is below the detection limit of conventional immuno-blotting methods. We further developed image analyses to quantify GSDMD pore dynamics and examine pore shape in lieu of direct super-resolution observations. To our knowledge, the dynamics of large membrane pores have not been captured in this manner. We resolved the spatiotemporal reach of the GSDMD-calcium "compartments": these 10 μm zones may inhibit the function of neighboring pores during the open time span. Comparison between electrophysiology and live-cell biosensing confirmed that on average, GSDMD pores open for tens of seconds (Fig. 3b and Fig. 5c, respectively). We presume that faster events (Supplementary Fig. 5) carry lower calcium fluxes that may not transmit substantial signals to endogenous enzymes. Intriguingly, neither live-cell biosensing nor electrophysiology showed gross disruption in calcium flux or ion current, suggesting that pore closure occurred cleanly and remained approximately at the original location. We surmise that this is possible through an ellipsoidal, "eye"-shaped intermediate that could spring back-and-forth cooperatively to accommodate the lateral stability of the phospholipid bilayer (Fig. 6e, schematic). This

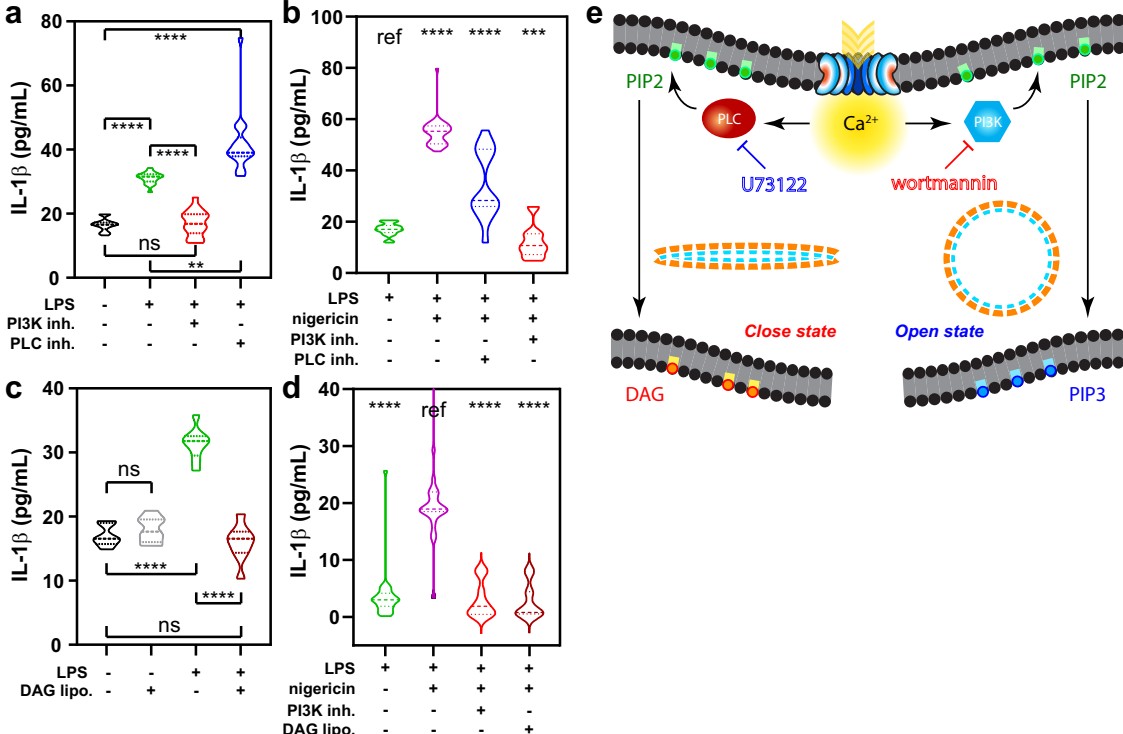

**Fig. 6 Inflammatory IL-1β cytokine release from macrophages can be controlled by altering the balance of the calcium-phosphoinositide circuit.** IL-1β release was dampened by wortmannin but enhanced by U73122. This is shown for both **a** short-term LPS stimulation and **b** LPS-primed inflammasome activation via nigericin. Alternatively, diacylglycerol (DAG)-rich liposomes were used to dampen IL-1β, also performed in **c** short-term LPS stimulation and **d** LPS-primed inflammasome activation via nigericin. Each dataset was obtained from ELISA measurements over a number of technical replicates (tr) of bone marrow-derived macrophages (BMDM) from separate mice (n). For (**a**) n = 3, tr = 8; for (**b**) n = 5, tr = 8; for (**c**) n = 3, tr = 8; and for (**d**) n = 4, tr = 8. For panels (**a**) to (**d**), statistical significance was reported from Brown–Forsythe and Welch ANOVA with multiple comparisons; "ref" represents the reference; in all instances ****$p < 0.0001$; ns not significant. In (**a**), **$p = 0.0011$; in (**b**), ***$p = 0.0003$. **e** Summary scheme of the proposed calcium-driven self-regulation of GSDMD pore dynamics.

conformational change is parsimonious for repeated, fast regulation of the oligomeric pore. This concept is further supported by atomic force microscopy observation of stable, arc-shaped GSDMD oligomers[6]. We observed a wide distribution of peak single pore currents compared to that typical of "unitary" single ion channels. Since peak current is directly correlated to the opened-pore size, we suspect that at least two factors are at play. First, the ellipsoidal model operates in a contiguous membrane environment, with an "analog" freedom to shut without opening fully into a circle. Since static structures showed that the oligomeric size of gasdermin pores was not always the same[6,35], we also surmise that activated monomers dynamically exit and re-join, thereby modulating the maximum pore size around a mean value. Complications of live-cell imaging (such as lack of exact flare geometry, occluded symmetry) prevented the use of fitting/localization algorithm on these datasets to determine whether the pore centroids were displaced below the diffraction limit over time. Since the calcium flares did not present the continuous structures required to enable ultra-fast super-resolution approaches[47], novel ultra-fast super-resolution methods will likely be required to directly visualize these intermediate states. Together, these data show that GSDMD pores have distinct dynamics, in contrast to unitary ion channels that show constant size and currents.

Whereas upstream inflammatory caspase regulation contributes to the generation of active gasdermins[48], the present study shows that membrane-gasdermin dynamics is also a major layer of intrinsic regulation in pyroptosis. We showed that a calcium-driven phosphoinositide bias towards PIP3 strengthened pore stability and drove cytokine release and inflammation. This mechanistically unifies disparate recent findings where

magnesium blocked GSDMD-induced pyroptosis;[49] PI3K inhibition prevented inflammatory IL-1β in mouse macrophages;[50] and PLC inhibition in a mouse peritonitis model significantly reduced macrophage and lymphocyte numbers[51] and increased inflammatory IL-6 production in human monocytes[52]. The mechanism described in the present study was supported by two distinct, short- and long-term inflammatory cytokine release models in living bone marrow cells derived from mouse macrophages. The results also suggested that DAG and its functional mimics may be explored to control cytokine release, at least for GSDMD. It is possible that other gasdermin family members[1] display dynamic conformations that are similarly regulated by their lipid microenvironment, which are fascinating avenues for future work. In summary, we showed that oligomeric GSDMD pore dynamics mediate an intricate, balanced calcium signaling to drive the opening or closing of pores and can thereby self-regulate pyroptosis.

## Methods

**Protein Purification.** Plasmid encoding the full-length human GSDMD with an N-terminal His-SUMO tag was transformed into *E. Coli* BL21 strain. The cultures were grown at 37 °C until OD600 0.8, and protein expression was induced with 0.2 mM IPTG overnight at 20 °C. The cells were harvested by centrifugation and the pellet was resuspended in 20 mM Tris buffer pH 7.5, 50 mM NaCl, 5 mM imidazole, 20 mM MgCl2, 10 mM KCl, 0.5 mM TCEP, 0.1 mM protease inhibitor, and DNase I. After resuspension, cells were disrupted by high-pressure and centrifuged at 30,000 × g at 4 °C for 1 h. The supernatant was incubated for 2 h at room temperature with pre-equilibrated Ni-NTA affinity resin (Thermo Scientific) and then passed through a column for gravity flow purification. The column was washed with 20 column volumes of resuspension buffer, and the fusion protein was eluted with 3 column volumes of the same buffer with 250 mM imidazole. SUMO-

tag cleavage was achieved by the addition of ULP1 protease to the solution and subsequent dialysis overnight at 4 °C against 20 mM Tris buffer pH 7.5, 50 mM NaCl, 0.5 mM TCEP. GSDMD was eluted from the second round of purification through pre-equilibrated Ni-NTA affinity resin. The protein was further purified by Hi-trapQ ion-exchange and a Superdex 75 gel filtration column (GE Healthcare) pre-equilibrated with 20 mM Tris buffer pH 7.5, 50 mM NaCl, 0.5 mM TCEP. The purified protein was concentrated to 20 mg/ml and frozen at −80 °C.

**Free-standing lipid bilayer formation and electrophysiology**. We employed a bilayer chamber with a polystyrene cuvette. Using the "folding" method based on Langmuir–Blodgett film[53,54], monolayers are spread in the air–water interface at the chamber pinhole[55,56] via a shallow polystyrene trough filled with recording solution. Bilayers were formed by two monolayers of 1:3 POPE/POPC (Avanti-lipids) in pentane; where appropriate, phosphoinositides of the indicated head-group were introduced with di-oleoyl (18:1) fatty acid chains at 10% relative concentration. A small amount of phospholipid-dispersed in pentane solution is dropped on one of the water surfaces and the lipid molecules remain at the air–water interface as the solvent evaporates, forcing the lipid molecules to form a monolayer; this is repeated at the air–water interface in each compartment. Then, by adding more solution and therefore raising the recording solution level, the monolayer is lifted up the cuvette surface and passes through the pinhole, where the two monolayers from either side meet and form a bilayer. The membrane formed by this method is referred to as "solvent-free", with an advantage over others such as the "painting" method in its ability to form an asymmetrical membrane where the lipid composition of either leaflet is arbitrarily defined[57]. Once formed, the membrane largely retains this asymmetry as the flip-flop exchange of lipids across the leaflets is very slow.

For incorporation of the GSDMD into the lipid bilayer, 2 μM of the purified GSDMD protein as well as 2 μM of recombinant caspase-1 was added to the cis side chamber. The specific membrane capacitance of the membrane formed by the folding method is 0.6–0.8 μF/cm$^2$. This value is close to the native biological membrane, suggesting that practically no solvent layer exists between the two monolayers. After the formation of a stable lipid bilayer and after the addition of GSDMD and caspase-1 mix by perfusion system, the lipid bilayer was clamped at 100 mV using a gap-free protocol, using an Axopatch 200B amplifier (Molecular Devices) with a Digidata 1440 A (Molecular Devices) to record while applying constant voltages to the lipid bilayer. The *cis* solution consists of 135 mM CsSO$_3$CH$_3$, 8 mM NaCl, 2 mM MgCl$_2$, 0.5 mM CaCl$_2$, 2 mM EGTA, and 10 mM HEPES at pH 7.2; the *trans* solution consists of 145 mM NaCl2, 2 mM CaCl$_2$, 1 mM MgCl$_2$, and 10 mM HEPES at pH 7.4. Currents through the voltage-clamped bilayers (background conductance <3 pS) are filtered at the amplifier output (via a low-pass filter of −3 dB at 10 kHz followed by 8-pole Bessel response). Data were secondarily high-pass filtered at 100 Hz, digitized at 1 kHz, and subsequently analyzed, using Clampfit (Clampex 11.0) and scripts written in R (version 4.0.3). The recording pipette was filled with an internal solution. For controls, only GSDMD or caspase-1 alone were added to the system. For the phosphoinositide incorporation, the same lipid bilayer formation technique was utilized with different phosphoinositides at the cis side of the chamber before the bilayer was formed.

To record whole-cell currents in small lung endothelial cells, cultured cells were patched at room temperature with an Axopatch 200B patch-clamp amplifier controlled via a Digidata 1440 A (Molecular Devices). Patch pipettes of 2 to 5 MΩ contained 156 mM CsCl, 1 mM MgCl$_2$, 10 mM CaCl$_2$, 10 mM EGTA, and 10 mM HEPES, yielding 10 μM free calcium (calculated with Webmaxc Standard, https://web.stanford.edu/~cpatton/webmaxcS.htm) (pH 7.4) and with a preincubated mix of recombinant GSDMD and caspase-1. The saline bath solution contained 140 mM NaCl, 4.8 mM KCl, 1.2 mM MgCl$_2$, 2.0 mM CaCl$_2$, 10 mM glucose, and 10 mM HEPES (pH 7.4). Cells were held at 0 mV, and 200 ms ramps from −100 to 100 mV were applied every 2 s. Currents were digitized at 10 kHz and low-pass-filtered at 2 kHz.

We observe two categories of events: formal ion flow current, and also minute, microscopic current. We have interpreted the 0.1–0.5 pA microscopic fluctuations, which invariably occurs once per experiment following protein introduction and always precedes large conducting currents, as dielectric micro-current indicating protein–membrane interactions. The steps for macroscopic ion flow current calculations using Clampfit are briefly outlined below. We approach event searches in three different ways. First, we idealize the data using a single pore search algorithm, which superimposes a best-fit square wave over each channel opening, translating single-channel records into idealized events that were then categorized using "Event Detection/Single-Channel Search". We also performed template searches by averaging trace segments that were manually identified single pore events. These templates were further utilized to identify events. Thirdly, threshold-based searches rely on amplitude baseline, marking events which crosses the thresholds. Using these identified events, we performed a P(open) analysis that represents the probability that a channel is open. This analysis workflow results in the reported pore characteristics, which includes (a) the number of single pore events, (b) the total time range, (c) dwell time (open duration), and (d) the probability of a single pore being open (open probability vs. dwell time).

**Cloning**. Cloning and subcloning were performed using the DH5α strain of *E. coli*. All mammalian constructs were cloned into the pcDNA3.0 vector with a modified multiple cloning site. Plasmids were generated as follows. Human Gasdermin D and PhoCl genes were obtained from Addgene (Plasmids #111559 and 87693), gifts from Drs. Hongbo Luo and Robert Campbell, respectively. PhoDer was constructed according to the reported domains structure by ligating three fragments into the pcDNA3.0 vector in one reaction via Gibson Assembly (E2611S, New England BioLabs). For each fragment, PCR was specifically designed to create appropriate complementation. jRCaMP1b gene was obtained from Addgene (Plasmid #100851). Membrane-targeted calcium indicator jRCaMP1b was constructed by ligating a PCR fragment of the calcium indicator into the restriction enzyme sites BamHI and EcoRI in a modified pcDNA3.0 vector carrying a 5′ fragment encoding the N-terminal Lyn kinase localization sequence. Site-directed mutagenesis was carried out via a previously published protocol[58,59] using mutagenic primers the chosen residues via E. coli DH5α strain; mutant residue numbering followed the published sequence of hGSDMD in NCBI database (https://www.ncbi.nlm.nih.gov/CCDS/CcdsBrowse.cgi?REQUEST=GENEID&DATA=79792).

**Cell culture**. The HeLa and RAW264.7 cells were maintained in DMEM media supplemented with 10% FBS and 1% penicillin and streptomycin. Cells were transfected at 50–70% confluency using Lipofectamine 2000 and incubated for 24 h before imaging. To examine endogenous inflammatory activation, RAW264.7 cells were treated with 100 ng/μL LPS (Invivogen) at the beginning of each experiment. Pretreatment to bias membrane composition by PIP2 modifying enzymes was performed by incubating HeLa cells in HBSS buffer with inhibitors at 37 °C for 10 min prior to imaging at the following concentrations: for PI3K inhibition: cells were treated with 10 μM wortmannin (Tocris); for PLC inhibition: cells were treated with 10 μM U73122 (Tocris). Similarly, for PLC activation: cells were treated with 50 μM m-3m3FBS (or negative control o-3m3FBS, both Tocris) for 10 min; for PI3K activation: cells were treated with 50 μg/mL 740 Y-P (Tocris) for 20 min. Cells were removed from the incubator and allowed to come to thermal equilibrium before being imaged in HBSS buffer at room temperature.

Bone marrow-derived macrophages were isolated using a modified version of the published protocol[60]. In brief, C57BL/6 J mice were initially acquired from The Jackson Laboratory, housed and bred at the University of Illinois at Chicago (UIC) in accordance with institutional and NIH guidelines. Female and male mice, kept at ambient temperature 18–23 °C (65–75 °F) and humidity 40–60% with water and food accessibility at all times in UIC animal facility, aged between 4–6 months, were randomized, sacrificed, and legs were removed. Leg tissues were cleaned to expose only the femoral bone, and the end of the bone was cut off to expose the bone marrow, which was then expelled from the bone with a jet of media using a 27 g needle into an Eppendorf tube. The cells were then resuspended in an L929 conditioned medium, whose major active component is M-CSF, in a 10 cm dish and incubated overnight. The medium was changed the next day to remove dead cells and changed every other day until cells were ready to use. Animal protocols were approved by the Office of Animal Care and Institutional Biosafety Committees, the University of Illinois at Chicago.

**Fluorescence imaging, optogenetic activation, and post-processing**. All epi-fluorescence imaging was performed on a customized Nikon Ti2-E microscope, operated via Nikon Elements software ver. 5.11.01, equipped with individual control over excitation (EX) and emission (EM) filters, a fast LED light source (Lumencor) with fluid-cooled EMCCD (Andor), motorized stage, and examined under a 40X or 100X oil immersion objective under live focus tracking. The following filters were used for RFP was performed using the following excitation (EX) and emission (EM) filter combinations (maxima/bandwidth in nm). Cellular fluorescence was examined at 12–15 s intervals with the exception of data presented in Supplementary Fig. 5, which were monitored at 1 s interval to capture faster calcium dynamics. A green fluorescence channel was utilized to monitor PhoDer expression in each cell due to the color of PhoCl. Emission intensities of individual cells were background-subtracted before normalization to zero time point when optogenetic activation began. In our optical setup, each round of PhoDer optogenetic activation (with either wildtype or mutant GSDMD) was performed using an EX filter at 440/20 nm for 20–30 ms across whole fields of view. Continuous PhoDer activation through 395/25 nm EX filter led to a fast activation of GSDMD, causing rapid cell "bubbling", blebbing and death in a calcium response time course comparable to nigericin treatment (Supplementary Fig. 1b, right panels). Where applicable, cells were treated with inhibitors (U73122, wortmannin) at the indicated concentration for 10 min prior to imaging. Likewise, cells were treated with activators and their control (m-3m3FBS, o-3m3FBS, and 740 Y-P) at the indicated concentration prior to imaging.

To capture the coincident calcium and dye flares, we utilized previously reported bright, turn-on malachite green (MG) binding aptamer (MGA)[61,62] and fluorogen activating protein (FAP)[11] in combination so both the membrane and cytosol of HeLa cells become more fluorescent upon MG influx. HeLa cells co-expressing PhoDer, Lyn-jRCaMP1b, and a construct carrying both Lyn-FAP and MGA aptamer repeats 3′ of the stop codon of the nucleotide sequence encoding Lyn-FAP, were subjected to typical optogenetic activation parameters. MG was added to the medium at 1 μM where nonspecific fluorescence is known to be low;[63] EM filter 705/72 was utilized for far-red channel MG fluorescence. The far-red images series were then post-processed via in-house Matlab (2019a/b) program (based on *diff* function) to generate a per-pixel level time-differential image series

representing the flux detected by accumulated MG over time; a rolling-window average (three image frames) was applied prior to differentiation to decrease noise inherent in these discrete numerical operations. MG fluorescence is not known to be calcium sensitive. Coincidental MG flux did not accompany every calcium flare and presumably requires a particularly optimal combination of Lyn-FAP/MGA6 expression, GSDMD pore dynamics, and the larger dye molecule kinetics.

**Local calcium fluctuation analysis and quantification**. Calcium influx was utilized to clarify the geometry of the underlying pore, facilitating automated isolation and quantification of calcium flares in complex, live-cell conditions. A post-processing algorithm was developed based on the principles and normalization scheme detailed in previous works. Briefly, photochromic Stochastic Optical Fluctuation Imaging (pcSOFI)[30] calculation treats the calcium fluctuation from the flares in time-averaged series to generate high autocorrelation signals with high contrast at locales of high ion flux change. To truly reflect the local flux change quantitatively, the computed fluctuation value is further normalized to remove dependence on the number of biosensors reporting at the given location in a scheme detailed previously[31]. Thus, using Matlab (2019a/b), rolling-window mean and rolling-window autocorrelation values were calculated for each pixel over each entire fluorescence time course to quantify calcium fluctuation pixel-wise. A range of window sizes were examined using examples where calcium flares are visually distinct, and the optimal window size of 3.25 min was determined for all analyses. This created high contrast images where flares are reliably isolated against an intracellular calcium backdrop since the rate of change of intracellular calcium is far slower compared to those of the flares and well-filtered by the window size. All fluctuation images were then verified manually against intensity time course to ensure bona fide flare detection; particularly strong and nonoverlapping flares were chosen for calculating the average kinetic parameters reported. Flare isolation was performed using automated ImageJ (version 1.52 v or newer) thresholding and particle analysis and recovered geometric parameters such as circularity. The workflow was automated and applied to all PhoDer datasets and subsequently summarized.

**Cytokine release assays**. Lipopolysaccharide (LPS) stimulated IL-1β cytokine release from bone marrow monocyte-derived macrophages (BMDM) was measured using the Quantikine ELISA kit MLB00C (R&D Systems). In the short-term protocol, cells were seeded in 96-well plates and pretreated for 10 min with medium control, 10 µM U73122 (PLC inhibitor), 10 µM wortmannin (PI3K inhibitor), 50 µM m-3m3FBS (PLC activator), 50 µg/mL 740 Y-P (PI3K activator), or a DAG-rich liposome preparation. About 100 ng/µL LPS (Invivogen) was then added to the cells for 5 min and the cell-free supernatants were recovered and subjected to the ELISA kit according to the manufacturer's instructions. The concentration of the samples in 96-well plates was measured via absorbance at 450 nm using an Epoch Microplate Spectrophotometer (Biotek Instruments). The OD values in six technical repeats each from a total of three mice ($n = 18$) were converted into the IL-1β levels reported via a power-law fit of a serial dilution calibration curve. In the NLRP3 inflammasome induction protocol, BMDM cells were primed with LPS for 3 h at 37 °C; the priming medium was then replaced with a normal culture medium containing 10 µg/ml nigericin for 30 min at 37 °C. IL-1β release in the medium was then measured by ELISA via the same procedure as described above.

Briefly, to prepare the DAG liposomes, 2.6 µM total phospholipids was dispensed in a glass test tube in an 80:10:10 ratio (PC-PS-DAG) and dried in a fume hood under a gentle stream of nitrogen and further speed-vac for 60 min under high vacuum. Dioleoylglycerol was used to match the fatty acid chain length of the other phospholipids. About 2.6 mL of room temperature (RT) HBS is then added to the dried lipids and incubated for 1 h at RT covered by parafilm, and then vortexed vigorously to completely resuspend the phospholipids. The glass tube is suspended over a bath sonicator at RT and sonicated until the appearance changes from milky to nearly clear indicating the formation of lamellar liposomes with little light scattering; a typical sonication time is 15 min. Liposomes are stored on ice until use, and 50 ng of total lipids (~100 liposomes) were used as a treatment for each macrophage culture well for 10 min on a "belly-dancer" shaker prior to inflammatory stimulation.

**Statistical methods**. The reported statistical significance between control and experimental datasets were the results of ANOVA or two-tailed, unpaired Welch's t-tests calculated at 95% confidence level using GraphPad Prism (Versions 8.4 and newer).

**Reporting Summary**. Further information on research design is available in the Nature Research Reporting Summary linked to this article.

## Data availability
The data supporting the findings of this study are available within the article and its supplementary information files. Additional information and relevant data will be available from the corresponding author upon reasonable request. The constructs described in this manuscript will be submitted to the nonprofit repository, Addgene, for scientific sharing. Source data are provided with this paper.

## Code availability
The code described in this manuscript is available through GitHub (garymolab/2021_phoder) and archived (https://doi.org/10.5281/zenodo.5703319).

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

## Acknowledgements

G.C.H.M. thanks Jalees Rehman for valuable discussions. This work is supported by P01HL060678, R01HL090152, R01HL152515 (A.B.M.), T32HL007820 (K.P.S.), and P01HL151327 (G.C.H.M. and A.B.M.).

## Author contributions

G.C.H.M. and A.B.M. conceived of the study. G.C.H.M. and A.B.S.C.G. designed all experiments. A.B.S.C.G. performed the in vitro electrophysiology experiments, cytokine assay, and wrote programs for all related analyses. G.C.H.M. designed and made the optogenetic constructs, performed live-cell fluorescence imaging, and wrote programs for all related analyses. K.P.S. performed the live-cell fluorescence imaging for activator drugs, additional mutants, higher frequency calcium, and coincident calcium and large-molecule flux. G.C.H.M., A.B.S.C.G., and A.B.M. wrote the paper.

## Competing interests

The authors declare no competing interests.
