## [Peer Review File · Nature Communications]

Gasdermin D Pores Are Dynamically Regulated by Local Phosphoinositide CircuitryREVIEWER COMMENTS

Reviewer #1 (Remarks to the Author):

In the manuscript entitled “Gasdermin D Pores Are Dynamically Regulated by Local Phosphoinositide Circuitry” by Ana Beatriz Santa Cruz Garcia et al, the authors address the complex mechanisms by which the Gasdermin D (GSDMD) pores interact with target membranes, create conducting pathways, and control the transmembrane transport of physiologically relevant ions and large molecules.

The GSDMD pores have emerged as a major conduit for interleukin (IL) secretion from the cytosol; they may also induce pyroptosis, a form of lytic cell death. Therefore, the presented work is of outmost importance for a better understanding of inflammasome biology and pyroptosis. To achieve their scientific goals, the authors employed a wide variety of molecular biology and biophysical techniques, appropriate for the presented studies. They utilized both live cells and artificial membrane systems in combination with optogenetics, fluorescence biosensing and imaging, image analysis, electrophysiology, and ELISA to identify components of the intricate pathways responsible for pore formation into target membranes, regulation, and intracellular content release. The authors identified Ca²⁺ kinetics in live cells (resembling inflammatory stimulation), transient ionic currents in artificial and natural membranes containing GSDMD pores, influence of lipid composition on pore formation and biological activity, and influence of inhibitors on cytokine release. All the experimental data presented in this paper point out beyond any doubt the complex modulation of the GSDMD pore’s biological activity by numerous biochemical cues that imply Ca²⁺ ions, metabolically relevant pathways, or lipid composition. I consider that these are notable achievements presented in this paper.

Some of the sections require several clarifications with regards to methodology, data interpretation, and scientific claims. In vitro electrophysiology experiments were performed by using a classical approach, which employed a folded, asymmetrical bilayer lipid membranes and voltage-clamp measurements with the Axopatch200B amplifier. The method section indicates that the bilayer was clamped at various voltages ranging from -100 mV to 100 mV using the gap-free protocol. The voltage values utilized for I/V plots are obvious in Supplementary Figure 4 (obtained with the ramp protocol) but there is no indication of the voltages employed for the experiments described in Figures 2-3. In the same line, it is not clear what sampling time and filter were used to record these data, and this may prevent a proper assessment of their validity. Ramp range, sampling time, and filter values are provided only for the whole-cell experiments; the voltage used to achieve the data presented in Supplementary Figure 3B is not specified so the whole cell and bilayer single channel ionic currents may not be compared. The first three traces (upper) in Figure 3B show the absence of ionic currents through the bare POPC/POPE bilayer membrane, and also upon addition of either GSDMD or Caspase 1. The main text (Results section) indicates n=20 bilayers for these traces. Are each of these traces representing an average from 20 bilayers? I do not know if the same ionic current scale was used for the traces shown in Figure 2B, but it seems like the noise increased considerably in the presence of GSDMD, Caspase 1, and their combination compared with the pristine membrane. What would be the reason for such a noise increase?

The authors state that after addition of activated GSDMD, they detected “protein insertions into the bilayers characterized via gating micro-currents ($n = 20$ bilayers, Fig. 2B, highlighted region).” Is this trace a typical one, or an average of 20 traces? I do not understand how the highlighted region indicates protein insertion. All I see is a very small change in the amplitude of the ionic currents (~ 0.05 pA), which fluctuates up and down. This does not look like pore insertion/formation (the amplitudes of the ionic currents would be bigger, as shown in panel C). Is this symmetrical perturbation produced by the insertion of a single protein monomer into the bilayer? Current models suggest that a formed prepore binds the membrane prior to insertion; hence, insertion should lead to large conducting pathways into the membrane. How may one be sure that this small change in the ionic currents is indicative of true insertions? Why do the ionic currents go up and down upon protein insertion?

Figure 2C shows what would be anticipated from pore insertion into a lipid membrane, i.e. larger ionic currents. It is clear that the ionic currents are transient, typical to ON/OFF states. It is striking that the amplitude of the currents is very non-uniform, which is quite different from the behavior of other ion channels or pore-forming proteins. Some of the events seem to be very short; an improper sampling rate/filter may reduce the amplitude of the events, and that is why it is important to specify these values in the methods section. For longer events, one may observe that the amplitude of the open current may change significantly in time. The authors assume a change in pore’s geometry (circular-elliptical). They may also consider that incomplete pores (i.e., arc-shaped) gain or lose one or more monomers and adjust the conductance. What was the voltage used for this experiment? Judging from the mean current and the results shown in Supplementary Figure 4 (I/V plot), it seems like the voltage was somewhere between 30 mV and 40 mV if a single pore was reconstituted into the membrane. In relation to these results, Supplementary Figure 4 shows a non-linear I/V curve ($n = 9$), and the authors state that the plot demonstrates dual-rectification and lack of selectivity. This requires a more thorough analysis. How many pores were reconstituted into the bilayer membrane for this assessment? A correct analysis of the I/V plot may not be made since the number of reconstituted pores is not known, and the voltage used to achieve the results shown in Figure 2C is not specified. The pores may present a voltage-dependent open probability, and this is not sufficiently addressed in the experimental section. The pore may prefer a closed state in the -30 mV: $+30$ mV voltage range, which may explain the non-linear I/V plot. Also, there is a clear decrease of the amplitude of the ionic currents manifesting at large hyperpolarization (~ -100 mV), which is not observed at large depolarizations ($\sim +100$ mV). In my opinion, this begs for assessing the voltage dependency of the open probability for an extended voltage range.

The interpretation of the results shown in Supplementary Figure 4 is also impeded by the lack of experimental details with regards to the recording protocol. Only the voltage range is obvious from the figure. What was the time length of the ramp? This parameter may be essential for data accuracy and interpretation. A fast ramp may introduce artifacts owing to a large capacitive current. Also, slow activation/deactivation of the pores in response to variable voltage may prevent achieving steady-state, and hinder measuring the ionic currents at equilibrium. Figure 2C shows a fluctuating ionic current, but the I/V plot (Supplementary Figure 4) is very smooth. I understand that this I/V plot is an averaged trace ($n=9$); were transient currents recorded in the individual traces? Is there any evidence that the voltage modulates the ON-OFF transitions? How does the IV plot change upon adjusting the length of the voltage ramp (scan rate)?

Additional experimental and methodology details are also needed for the section describing the phosphoinositide dependency by introducing PIP2, DAG, or PIP3; the results are shown in Figure 3. The

previous comments with regards to used voltage, sampling rate, filtering, and identification of protein insertion should be addressed as well. I do not know what the open duration of the pore (Figure 3b) represents and how it was calculated. I clearly observe faster ON/OFF transients for the PIP2 membrane (excessive filtering might constitute one of the causes for the reduced amplitudes), but I do not see such flickering for the PIP3 case. While I see a transient behavior, it does not look like open/close transitions, hence I am not confident on the reported single channel current value. It might be a single, evolving pore in the membrane, which does not close at all. I am also confused on what Figure 3C represents. It shows Open probability as a function of Dwell time for three different membrane compositions. How was this open probability calculated, and at what voltage? Customarily, dwell time is used to determine the open probability, and I do not understand what the presented distribution of the open probability as a function of dwell times shows. Clarifications are needed to better understand what this panel indicates, and addition of experimental estimations of open probability as a function of voltage is desirable.

PIP replacement with DAG in the bilayer membrane led to the absence of any open/close events in the trace (Figure 3E), yet it is claimed that the protein inserts into the target membrane (Supplementary Figure 5). This figure shows a similar pattern to Figure 2B. The authors must detail how these experimental data are interpreted to support the claim that they indicate true insertion events. I do not see any significant variation of the open current, and the increased noise may have a different origin. The claim that “the presence of DAG induced the closed state” is not necessarily true since there is no sufficient evidence that a functional pore was reconstituted in such a membrane. In the same context, the authors declare that PIP3 content kept the pores predominantly in the open state. Figure 3 shows a larger open probability for the PIP3 membrane at shorter dwell times (panel C), but panel B indicates the shortest open duration for the same membrane. This is not necessarily a contradiction but necessitates a detailed description of how the open probability and open duration are determined. This is imperative for PIP3 since it is difficult to identify any closing of the pore in the trace shown in Figure 3E.

Based on all the evidence presented in this work, there is no doubt that the GSDMD pores undergo complex regulation in both artificial and natural membrane systems but I am unsure of the validity of the proposed closing mechanism (i.e., an eye-shaped intermediate that springs back and forth). The very large opening of the β -barrel pore (~20 nm) and the apparent absence of moving parts seriously impede envisioning the closing mechanism responsible for regulation. Previous reports (referenced in this work) indicate that GSDMD may ensemble into a large variety of intermediates (rings, arc-shaped, and slits). This is quite similar to pore-forming toxins (i.e., Streptolysin), which I am not aware of being endowed with clear conductance regulation mechanisms. All previous structural data on GSDMD pores (including AFM, ref. 5) are rather static and the evolution of the pores was not sufficiently assessed in prior work. This model of oligomerization and pore formation (ref. 5) suggests that the ring-shaped oligomers are the most stable, which is anticipated, and that the intermediate arc- and slit-shaped oligomers evolve into rings with time. The same structural data clearly shows elliptic pores, but they seem to be incompletely formed. While the proposed regulatory model presented in this manuscript is very appealing, I have doubts that a complete ring structure (as shown in Figure 6E), which is rigid and stable, may undergo such dramatic conformational changes. Since the intermediate structures may also ensure transmembrane transport, the biochemical cues and circuitries identified in this work may also favor oligomerization into particular shapes/intermediates. The unstable intermediates may be responsible

for transient responses, while the stable pores, once formed, may lead to cellular death. The discussion section suggests that the eye-shaped intermediates are actually regulated, but this section may be expanded to provide more details and alternative regulatory pathways.

I consider that in some instances the clarity of the scientific message could be improved by avoiding using terms that are too broad for the context. For example, pore dynamics is much broader than pore regulation; I would recommend using regulation when describing the ON/OFF conductance changes. In the same line, terms like “finite kinetics”, “enhanced conformation flexibility”, “different conformational landscape”, “reduced pore activity dynamics”, or “action of the pores” may benefit from a better explanation of their meaning within the context—they are also too broad for the reader to precisely understand what specific functionality is described.

The authors may significantly improve this work by providing more details on the methodological aspects presented in this paper, expanding the electrophysiology investigations on planar membranes for providing insights into potential voltage-induced regulation, and providing alternatives for the claims not fully supported by this work. I believe that this will not only make the work more appealing to scientists from different backgrounds, but will also help with the reproducibility of the experimental work by other interested groups.

Reviewer #2 (Remarks to the Author):

The authors studied whether and how the channel activity of activated GSDMD, which forms a large pore, is regulated. By creatively inserting the light-cleavable PhoDer domain between the pore-forming GSDMD N-terminal and the autoinhibitory C-terminal, they were able to optogenetically activate the GSDMD pore-forming capability. The authors then combined the optogenetic approach with Ca²⁺-imaging and electrophysiology, and convincingly demonstrated that the forming and opening/closing of GSDMD pores are regulated by calcium-dependent modification of the membrane lipid composition. They specifically showed that the channel activity of GSDMD pores is facilitated by PIP₂ and PIP₃ and blocked by DAG, which in many ways is reminiscent of the regulations of some classic channels, like TRP channels, TREK1, and KCNQ1. These observations are surprisingly opposite to the intuitive expectation that the large pores of GSDMD might remain constantly open. More importantly, they may provide a mechanistically unified explanation on several recent observations that GSDMD pore formation does not necessarily lead to pyroptosis, and Mg²⁺ blocks GSDMD-induced pyroptosis, and that inhibiting PI3K and inhibiting PLC have opposite effects on GSDMD-dependent cytokine release.

Nevertheless, the authors should address some apparently inconsistencies and clarify some of the quite vague discussions.

(1) In Figure S1 and S2A, the authors measured calcium responses in macrophages stimulated with extracellular LPS. The features of these responses were subjectively assumed to be properties of GSDMD pores formed under physiological conditions, which was then used as the benchmark for the optogenetic GSDMD pore reports developed later in this study. However, there is a serious flaw here. How do the authors know that the observed calcium responses after extracellular LPS stimulation directly result from GSDMD pore formation, particularly in the absence of any negative control by genetic manipulation?

(2) Imaging reveals the ON/OFF rates of individual flares at the scale of tens of seconds (Figure 1), yet electrophysiological recordings from whole cells and reconstituted lipid bilayers indicates much more rapid flickering (most at the scale of tens of miniseconds; Figures 2, 3 and S3). Why are they so different?

(3) I am surprised by the strong voltage dependency of the GSDMD channels. However, they are not very consistent. In whole cells, they exhibit striking outward rectification, with clear opening $>+10$ mV (Figure S3A). In lipid bilayers, they show dual rectification. How to explain this difference? Moreover, it would help if the authors can demonstrate that living cells with functional GSDMD channels have a very depolarized membrane potential relevant to the very depolarized gating voltage.

(4) Importantly, this voltage dependency introduces another complicating factor into the membrane phosphoinositide dependency. Do PIP₂, PIP₃, and DAG change the voltage dependency of GSDMD channels? This should be tested experimentally.

(5) The authors proposed an interesting model that, unlike classic unitary channels, the large GSDMD pores may form ellipsoidal, "eye"-shaped states that are dynamically controlled by phosphoinositide compositions in the membrane. I like this model, but it also leads to the paradox when using the term "single channel". We often define "single channels" based on the observations of unitary currents. Here they are not very unitary, and in some striking cases (Figure 3E), defies such standard in every possible way. In addition, DAG applications not only strongly reduce the current amplitude but also protein insertion, suggesting that the putative "ellipsoidal" pore might not only be collapsed but also tentatively removed from the membrane. Therefore, the authors should make efforts in clarifying this term.

(6) Is it possible to test the effect of PI3K activators and PLC activators, in addition to the inhibitors?

(7) There are lots of serious issues in reference citation. Reference 6 and 7 are the two original publications reporting GSDMD cleavage by caspase-1 and -11 and the induction of pyroptosis, but the latter one was not cited together reference 6 in the Introduction session. PMID: 27281216 is the most comprehensive original publication that describes the pore-forming activity of the gasdermin family

including GSDMD cleaved by caspase-1/11, but this reference was omitted (while other much less elegant papers were all cited) although the key mutations (affecting phosphoinositide binding) were deduced from the information reported in PMID: 27281216; when introducing the anti-tumor activity of gasdermin pore formation and pyroptosis, PMID: 32188939 and 32299851, the two most important publications were also omitted in place of citing reference 16. When introducing structural analyses of gasdermins (references 30 and 31), the authors also missed two more original and elegant publications (PMID: 27281216 and PMID: 32109412) that report the auto-inhibited gasdermin structure and complex structures between GSDMD and caspase-1 (as well as caspase-11 and -4). As for the concept of pyroptosis, PMID: 27932073 is the first perspective article that redefines pyroptosis as gasdermin-mediated programmed necrotic cell death and as usual this reference was missing in the relevant place in this article.

What does “may even requires apoptotic reinforcement to ensure cell death (reference 19) mean? I do not think reference 19 really has the intended meaning. Even though, PMID: 28459430 is the first reference elegantly showing that caspase-3, previously known as apoptotic caspase, can cleavage GSDME to drive the cells into pyroptosis, but unfortunately this reference was also omitted in the manuscript.

(8) Does the observed dynamics of pore opening/closing also apply to other gasdermins as gasdermin-B, C and E have also been shown to execute pyroptosis under important biological contexts? The reviewer is also curious whether cardiolipin, the more preferred lipid target of the gasdermin family, shows the similar property as phosphoinositide. They authors should make the effort to test this as the experiments are quite straightforward.

Minor concerns:

(1) In addition to the current amplitudes in pA, the authors should also report the conductance of putative single channels.

(2) In Methods, the authors described no Ca²⁺ in the bath solution for whole-cell recordings. Please check if this is true.

(3) The text needs to be more polished. There are some clear typos and errors. For example, in p8, the sentence "the enzymes PI3K and PLC promote the relative accumulation of diacylglycerol (DAG) ..." needs to swap the position of DAG and PIP3.

Point-by-point Response

Reviewer #1 (Remarks to the Author):

In the manuscript entitled “Gasdermin D Pores Are Dynamically Regulated by Local Phosphoinositide Circuitry” by Ana Beatriz Santa Cruz Garcia et al, the authors address the complex mechanisms by which the Gasdermin D (GSDMD) pores interact with target membranes, create conducting pathways, and control the transmembrane transport of physiologically relevant ions and large molecules.

The GSDMD pores have emerged as a major conduit for interleukin (IL) secretion from the cytosol; they may also induce pyroptosis, a form of lytic cell death. Therefore, the presented work is of utmost importance for a better understanding of inflammasome biology and pyroptosis. To achieve their scientific goals, the authors employed a wide variety of molecular biology and biophysical techniques, appropriate for the presented studies. They utilized both live cells and artificial membrane systems in combination with optogenetics, fluorescence biosensing and imaging, image analysis, electrophysiology, and ELISA to identify components of the intricate pathways responsible for pore formation into target membranes, regulation, and intracellular content release. The authors identified Ca^{2+} kinetics in live cells (resembling inflammatory stimulation), transient ionic currents in artificial and natural membranes containing GSDMD pores, influence of lipid composition on pore formation and biological activity, and influence of inhibitors on cytokine release. All the experimental data presented in this paper point out beyond any doubt the complex modulation of the GSDMD pore's biological activity by numerous biochemical cues that imply Ca^{2+} ions, metabolically relevant pathways, or lipid composition. I consider that these are notable achievements presented in this paper.

We are grateful for the reviewer's complementary remarks. We have addressed all points raised by the reviewer. However, as related points were raised throughout different comment sections, we will sometimes defer to a later point for a detailed response or point out comments that were already addressed.

Some of the sections require several clarifications with regards to methodology, data interpretation, and scientific claims. In vitro electrophysiology experiments were performed by using a classical approach, which employed a folded, asymmetrical bilayer lipid membranes and voltage-clamp measurements with the Axopatch200B amplifier. The method section indicates that the bilayer was clamped at various voltages ranging from -100 mV to 100 mV using the gap-free protocol. The voltage values utilized for I/V plots are obvious in Supplementary Figure 4 (obtained with the ramp protocol) but there is no indication of the voltages employed for the experiments described in Figures 2-3. In the same line, it is not clear what sampling time and filter were used to record these data, and this may prevent a proper assessment of their validity. Ramp range, sampling time, and filter values are provided only for the whole-cell experiments; the voltage used to achieve the data presented in Supplementary Figure 3B is not specified so the whole cell and bilayer single channel ionic currents may not be compared.

In response, we regret if the Supplementary Information was unclear. The gap-free experiments shown in Figs. 2-3 were performed by clamping the bilayer at 100 mV. Currents through the voltage-clamped bilayers (background conductance <3 pS) are low-pass filtered at the amplifier output (-3 dB at 10 kHz, 8-pole Bessel response), then high-pass filtered at 100 Hz and digitized at 1 kHz. The

ramp experiments were recorded with a 200 ms ramp protocol from -100 mV to 100 mV with a hold potential of 0 mV. We have now updated the Supplementary to include this information.

The reviewer is correct that the single pore currents in whole cell and bilayer models are not directly comparable. However, this is not caused by mismatch in electrophysiological parameters, but due to the inherent differences between the systems such as buffer conditions, membrane complexity, etc.

The first three traces (upper) in Figure 3B show the absence of ionic currents through the bare POPC/POPE bilayer membrane, and also upon addition of either GSDMD or Caspase 1. The main text (Results section) indicates $n=20$ bilayers for these traces. Are each of these traces representing an average from 20 bilayers? I do not know if the same ionic current scale was used for the traces shown in Figure 2B, but it seems like the noise increased considerably in the presence of GSDMD, Caspase 1, and their combination compared with the pristine membrane. What would be the reason for such a noise increase?

In response, averaging across experiments would mask the stochastic open/closing events that are key to our results. Thus, the traces are not averaged unless explicitly labeled; for electrophysiological data, only new Supplementary Fig. 5 (previously Supplementary Fig. 4) has been averaged.

We presume the reviewer is referring to “Fig. 2B” in the first part of the comment, as Fig. 3B does not match the description. Indeed, the same current scale has been utilized for all four traces shown in Fig. 2B. We believe the baseline “noise” increase represents the increased heterogeneity of the system upon the addition of protein components that can come into contact (but not binding) with the membrane bilayer.

The authors state that after addition of activated GSDMD, they detected “protein insertions into the bilayers characterized via gating micro-currents ($n = 20$ bilayers, Fig. 2B, highlighted region).” Is this trace a typical one, or an average of 20 traces? I do not understand how the highlighted region indicates protein insertion. All I see is a very small change in the amplitude of the ionic currents (~ 0.05 pA), which fluctuates up and down. This does not look like pore insertion/formation (the amplitudes of the ionic currents would be bigger, as shown in panel C). Is this symmetrical perturbation produced by the insertion of a single protein monomer into the bilayer? Current models suggest that a formed prepore binds the membrane prior to insertion; hence, insertion should lead to large conducting pathways into the membrane. How may one be sure that this small change in the ionic currents is indicative of true insertions? Why do the ionic currents go up and down upon protein insertion?

In response, the presented protein insertion traces are representative for the given condition. We observed these minute current events occurring once per experiment following protein introduction and always preceding large conducting currents – hence our interpretation that they signify protein insertions. We believe these small “gating currents” are dielectric current induced by disturbances upon protein entry, without formal ion flow, and hence they appear “up-and-down”.

A key conclusion of the data is the oligomeric nature of GSDMD pores yield a novel type of dynamism, which should not be compared with that of unitary ion channels. We made this point in the Discussion, and will further expand on this below. Here, we agree with the reviewer’s comment on monomer

entry/exit. It has indeed been demonstrated that active monomers can insert into the membrane (PMID: 29898893). This suggests that insertion may not be followed immediately by a large current because the pore need not insert as a complete unit capable of conducting ion flow.

Figure 2C shows what would be anticipated from pore insertion into a lipid membrane, i.e. larger ionic currents. It is clear that the ionic currents are transient, typical to ON/OFF states. It is striking that the amplitude of the currents is very non-uniform, which is quite different from the behavior of other ion channels or pore-forming proteins. Some of the events seem to be very short; an improper sampling rate/filter may reduce the amplitude of the events, and that is why it is important to specify these values in the methods section.

For longer events, one may observe that the amplitude of the open current may change significantly in time. The authors assume a change in pore's geometry (circular-elliptical). They may also consider that incomplete pores (i.e., arc-shaped) gain or lose one or more monomers and adjust the conductance. What was the voltage used for this experiment? Judging from the mean current and the results shown in Supplementary Figure 4 (I/V plot), it seems like the voltage was somewhere between 30 mV and 40 mV if a single pore was reconstituted into the membrane.

As mentioned above, we filtered the data using a low pass filter (-3 dB at 10 kHz), a high pass filter (100 Hz), then digitized at 1 kHz. This is a very common filtering practice (PMID: 23529424). We do not believe this will cause artifacts in our data. The clamping voltage used for these experiments were 100 mV.

We believe that oligomer symmetry of these pores is not precise; indeed, they have been found to vary (PMID: 29898893, 29695864). Thus, the number of constituent monomers may not be constant throughout the pore's lifetime. We believe our data suggest that the pore dynamics are the results of two main factors. Firstly, monomers can exit and re-join thereby altering the maximum pore size; second, the ellipsoidal pore may also not fully open to a circle before shutting. Therefore, the observation that peak currents are not uniform reflects how GSDMD pores are unlike static, unitary ion channels. As a consequence, they do not display integer uniform peak current amplitude, but vary in a wider distribution around a maximum, as shown in Fig. 3D.

We have now explicitly stated this conceptual ramification in the Results/Discussion sections.

In relation to these results, Supplementary Figure 4 shows a non-linear I/V curve (n = 9), and the authors state that the plot demonstrates dual-rectification and lack of selectivity. This requires a more thorough analysis. How many pores were reconstituted into the bilayer membrane for this assessment? A correct analysis of the I/V plot may not be made since the number of reconstituted pores is not known, and the voltage used to achieve the results shown in Figure 2C is not specified. The pores may present a voltage-dependent open probability, and this is not sufficiently addressed in the experimental section. The pore may prefer a closed state in the -30mV: +30 mV voltage range, which may explain the non-linear I/V plot. Also, there is a clear decrease of the amplitude of the ionic currents manifesting at large hyperpolarization (~-100 mV), which is not observed at large depolarizations (~+100 mV). In my opinion, this begs for assessing the voltage dependency of the open probability for an extended voltage range.

In response, the averaged curves (previously in Supplementary Fig. 4, now Supplementary Fig. 5) were recorded with a 200 ms ramp protocol from -100 mV to 100 mV with a hold potential of 0 mV. We do not know the precise number of pores from these total current ramp recordings, because as our gap free protocols show, their peak currents are non-uniform.

While the reviewer's comments on voltage dependence raises an interesting avenue for future studies, we believe it is beyond the scope of the present contribution. The voltage dependency of the open probability is not central to our demonstration that GSDMD pores are dynamic. For some ion channels, charge movement due to membrane (de)polarization drives conformational changes that in turn controls activity. In contrast, endogenous GSDMD activation requires caspase cleavage to begin working. Our data also directly showed that the subsequent signaling is driven by calcium, not membrane potential. Lastly, the dynamics of these pores show that they are not directly comparable to unitary ion channels. Thus, it seems likely that voltage dependence is not an intrinsic part of GSDMD physiology, as this alternative requires the membrane potential signal to work in synchrony with caspase upstream, and yet it might still be supplanted by calcium downstream. However, we have now included this possibility in Discussion to comply with the reviewer's suggestion to expand alternatives.

The interpretation of the results shown in Supplementary Figure 4 is also impeded by the lack of experimental details with regards to the recording protocol. Only the voltage range is obvious from the figure. What was the time length of the ramp? This parameter may be essential for data accuracy and interpretation. A fast ramp may introduce artifacts owing to a large capacitive current. Also, slow activation/deactivation of the pores in response to variable voltage may prevent achieving steady-state, and hinder measuring the ionic currents at equilibrium. Figure 2C shows a fluctuating ionic current, but the I/V plot (Supplementary Figure 4) is very smooth. I understand that this I/V plot is an averaged trace (n=9); were transient currents recorded in the individual traces? Is there any evidence that the voltage modulates the ON-OFF transitions? How does the IV plot change upon adjusting the length of the voltage ramp (scan rate)?

In response, as mentioned above, the averaged IV curves (old Supplementary Fig. 4; new Supplementary Fig. 5) were recorded with a ramp time length of 200 ms, from -100 mV to 100 mV with a hold potential of 0 mV. As the reviewer pointed out, sufficient time is needed to perform total current ramp recordings properly. Thus, scanning different lengths of voltage ramp may further complicate interpretation. This also means we could not detect fast transient currents in these ramps. Thus, we do not have evidence to suggest voltage dependence in ON-OFF transition.

Additional experimental and methodology details are also needed for the section describing the phosphoinositide dependency by introducing PIP2, DAG, or PIP3; the results are shown in Figure 3. The previous comments with regards to used voltage, sampling rate, filtering, and identification of protein insertion should be addressed as well. I do not know what the open duration of the pore (Figure 3b) represents and how it was calculated. I clearly observe faster ON/OFF transients for the PIP2 membrane (excessive filtering might constitute one of the causes for the reduced amplitudes), but I do not see such flickering for the PIP3 case. While I see a transient behavior, it does not look like open/close transitions, hence I am not confident on the reported single channel current value. It might be a single, evolving pore in the membrane, which does not close at all.

I am also confused on what Figure 3C represents. It shows Open probability as a function of Dwell time for three different membrane compositions. How was this open probability calculated, and at what voltage? Customarily, dwell time is used to determine the open probability, and I do not understand what the presented distribution of the open probability as a function of dwell times shows. Clarifications are needed to better understand what this panel indicates, and addition of experimental estimations of open probability as a function of voltage is desirable.

We regret any confusion. “Open duration” is the same as dwell time. We have relabeled Fig. 3B and updated the legend to minimize this issue.

The reviewer is correct that dwell time is used to determine open probability – this is why Fig. 3D is plotted to represent the change in open probability as a function of dwell time. As an example, Fig. 3C indicates the following: “in all registered single pore events, the probability of finding an event where the pore stayed open for ~750 ms is ~30%”.

PIP replacement with DAG in the bilayer membrane led to the absence of any open/close events in the trace (Figure 3E), yet it is claimed that the protein inserts into the target membrane (Supplementary Figure 5). This figure shows a similar pattern to Figure 2B. The authors must detail how these experimental data are interpreted to support the claim that they indicate true insertion events. I do not see any significant variation of the open current, and the increased noise may have a different origin. The claim that “the presence of DAG induced the closed state” is not necessarily true since there is no sufficient evidence that a functional pore was reconstituted in such a membrane. In the same context, the authors declare that PIP3 content kept the pores predominantly in the open state. Figure 3 shows a larger open probability for the PIP3 membrane at shorter dwell times (panel C), but panel B indicates the shortest open duration for the same membrane. This is not necessarily a contradiction but necessitates a detailed description of how the open probability and open duration are determined. This is imperative for PIP3 since it is difficult to identify any closing of the pore in the trace shown in Figure 3E.

We thank the reviewer for these helpful comments. We reiterate that our insertion traces represent “minute events occurring once per experiment following protein introduction and always precedes large conducting currents”. Across bilayer compositions, the dielectric “up/down” peaks in repeating waveforms of 0.1-0.5 pA magnitude are indeed similar, as the reviewer mentions. It supports the idea that they do indicate insertion events, as one would not expect the insertion of GSDMD N-terminal domains to be vastly different in each case. We have now clarified this in Supplementary Information.

Our claim that DAG induced the closed state, and vice versa, PIP3 induced the open state, is based on data from both electrophysiology and live cell calcium biosensing.

To calculate open duration and probability, we first identify the single pores events. We do so by 3 ways briefly outlined below. First, we translate single-pore records into idealized square waves. We also performed template searches by averaging trace segments that were manually identified single pore events. Thirdly, we use threshold-based searches rely to mark events which crosses the thresholds. These approaches allowed us to identify and capture the events, upon which we then perform P(open) analysis using Clampfit. This analysis workflow results in the single pore characteristics, which includes a) the number of events, b) the total time range, and c) the probability

of finding a single pore of given open duration in the entire time course (i.e. Fig. 3C). These details are now included in Supplementary Information.

We wash off excess activated GSDMD as soon as the first formal ion flow current occurs so subsequent events are not due to excess protein insertions. For DAG containing bilayers, whether we washed excess GSDMD had no impact – there were never large currents, only the minute dielectric insertion currents. For PIP3, the reviewer commented that “it is difficult to identify any closing of the pore”; this is consistent with our statement that PIP3 induced the open state. In Fig. 3B, PIP3 shows a short open duration/dwell time because those events that we could identify were short lived. The reason becomes clear when considering the representative trace in Fig. 3E: PIP3 containing bilayers experience increasing ion flow and osmotic pressure because the pore cannot shut (Fig. 3E, PIP3). Therefore, only a few short-lived single pore events could be registered while the current mounts. This was briefly stated in the main text.

We agree that since large macro-currents were not observed in bilayers containing DAG, we cannot be certain whether the inserted proteins formed functional pores. However, the concept that DAG content induces the close state is further consistent with that found in the live cell calcium signaling. A shift towards higher relative DAG content via wortmannin (PI3K inhibitor) treatment had the effect of delaying and slowing the single pores (Fig. 5A, right column, 5C/D), consistent with the idea that DAG content induces a closed state. Vice versa, when cells were treated with U72133 (PLC inhibitor) to shift the membrane to higher relative PIP3 content, cells rapidly accumulated calcium, suggesting that PIP3 content induces the opened state (Fig. 5A, middle column).

Based on all the evidence presented in this work, there is no doubt that the GSDMD pores undergo complex regulation in both artificial and natural membrane systems but I am unsure of the validity of the proposed closing mechanism (i.e., an eye-shaped intermediate that springs back and forth). The very large opening of the β -barrel pore (~ 20 nm) and the apparent absence of moving parts seriously impede envisioning the closing mechanism responsible for regulation. Previous reports (referenced in this work) indicate that GSDMD may ensemble into a large variety of intermediates (rings, arc-shaped, and slits). This is quite similar to pore-forming toxins (i.e., Streptolysin), which I am not aware of being endowed with clear conductance regulation mechanisms. All previous structural data on GSDMD pores (including AFM, ref. 5) are rather static and the evolution of the pores was not sufficiently assessed in prior work. This model of oligomerization and pore formation (ref. 5) suggests that the ring-shaped oligomers are the most stable, which is anticipated, and that the intermediate arc- and slit-shaped oligomers evolve into rings with time. The same structural data clearly shows elliptic pores, but they seem to be incompletely formed. While the proposed regulatory model presented in this manuscript is very appealing, I have doubts that a complete ring structure (as shown in Figure 6E), which is rigid and stable, may undergo such dramatic conformational changes.

We thank the reviewer for the encouraging comments. We wish to point out that circular structures such as that depicted in Fig. 6E, while compressively stable when pressed in from all directions, is not similarly rigid in tension. Thus, they are likely only as rigid/stable as their membrane support and subject to asymmetric instability along the ring. Our data specifically suggest that alterations in lipid composition are sufficient to affect the dynamics of these large oligomeric structures. We speculate that spontaneous lateral membrane pressure fluctuation, brought on by change in lipid

composition or dynamic monomer entry/exit, may serve as the catalyst to initiate a dramatic conformational change such as ellipsoidal closing.

Since the intermediate structures may also ensure transmembrane transport, the biochemical cues and circuitries identified in this work may also favor oligomerization into particular shapes/intermediates. The unstable intermediates may be responsible for transient responses, while the stable pores, once formed, may lead to cellular death. The discussion section suggests that the eye-shaped intermediates are actually regulated, but this section may be expanded to provide more details and alternative regulatory pathways.

In response, we have now included the above monomer entry/exit concept as part of the discussion on possible intermediary and alternatives.

I consider that in some instances the clarity of the scientific message could be improved by avoiding using terms that are too broad for the context. For example, pore dynamics is much broader than pore regulation; I would recommend using regulation when describing the ON/OFF conductance changes. In the same line, terms like “finite kinetics”, “enhanced conformation flexibility”, “different conformational landscape”, “reduced pore activity dynamics”, or “action of the pores” may benefit from a better explanation of their meaning within the context—they are also too broad for the reader to precisely understand what specific functionality is described.

In response, we have now removed descriptors such as “finite kinetics” and others to better clarify our model. As “regulation” could be confused with the key “membrane-GSDMD-calcium signaling regulation” described in the manuscript, we prefer to restrict the use of this term for these biochemical signals.

The authors may significantly improve this work by providing more details on the methodological aspects presented in this paper, expanding the electrophysiology investigations on planar membranes for providing insights into potential voltage-induced regulation, and providing alternatives for the claims not fully supported by this work. I believe that this will not only make the work more appealing to scientists from different backgrounds, but will also help with the reproducibility of the experimental work by other interested groups.

We thank the reviewer for the overall emphasis on methodological clarity. We have endeavored to improve our presentation by expanding details of the electrophysiology to cover the calculation methods as well as the recording parameters in the Supplementary. We have also incorporated some of the discussion points from this response into the main text to make the work more appealing, readable, and accessible.

Reviewer #2 (Remarks to the Author):

The authors studied whether and how the channel activity of activated GSDMD, which forms a large pore, is regulated. By creatively inserting the light-cleavable PhoDer domain between the pore-forming GSDMD N-terminal and the autoinhibitory C-terminal, they were able to optogenetically activate the GSDMD pore-forming capability. The authors then combined the optogenetic approach with Ca²⁺-imaging and electrophysiology, and convincingly demonstrated that the forming and opening/closing of GSDMD pores are regulated by calcium-dependent modification of the membrane lipid composition. They specifically showed that the channel activity of GSDMD pores is facilitated by PIP₂ and PIP₃ and blocked by DAG, which in many ways is reminiscent of the regulations of some classic channels, like TRP channels, TREK1, and KCNQ1. These observations are surprisingly opposite to the intuitive expectation that the large pores of GSDMD might remain constantly open. More importantly, they may provide a mechanistically unified explanation on several recent observations that GSDMD pore formation does not necessarily lead to pyroptosis, and Mg²⁺ blocks GSDMD-induced pyroptosis, and that inhibiting PI3K and inhibiting PLC have opposite effects on GSDMD-dependent cytokine release.

We thank the reviewer for the positive and encouraging comments.

Nevertheless, the authors should address some apparently inconsistencies and clarify some of the quite vague discussions.

(1) In Figure S1 and S2A, the authors measured calcium responses in macrophages stimulated with extracellular LPS. The features of these responses were subjectively assumed to be properties of GSDMD pores formed under physiological conditions, which was then used as the benchmark for the optogenetic GSDMD pore reports developed later in this study. However, there is a serious flaw here. How do the authors know that the observed calcium responses after extracellular LPS stimulation directly result from GSDMD pore formation, particularly in the absence of any negative control by genetic manipulation?

We apologize for the misunderstanding. While we have not yet ascertained whether the observed calcium responses were a direct result of GSDMD pore formation at the narrative juncture indicated by the reviewer, GSDMD activation is a well-known effect downstream of LPS (Fig. 2C in PMID: 33472215). Thus, Supplementary Figs. 1/2A (Fig. S1/S2A) are the impetus that drove the subsequent optogenetic design. This approach is the genetic manipulation: it is orthogonal to LPS signaling to avoid its divergent effects, and it can be selectively/controllably activated. Thus, we utilized optogenetics to suggest that the calcium response following LPS were the direct results of GSDMD activation, as optogenetic gasdermin activation alone could recreate both the kinetics and form of the LPS stimulated calcium response. We have updated the main text to reinforce this logic flow. We also include data to show that cells (side-by-side within the same field of view) do not display calcium responses if they lack PhoDer expression (Supplementary Fig. 2C).

(2) Imaging reveals the ON/OFF rates of individual flares at the scale of tens of seconds (Figure 1), yet electrophysiological recordings from whole cells and reconstituted lipid bilayers indicates much more rapid flickering (most at the scale of tens of miniseconds; Figures 2, 3 and S3). Why are they so

different?

We believe the differences between the time scale of the flares observed in live cells and the events recorded in bilayer models may not be directly comparable due to differences in the sensitivity of electrophysiology equipment and that of genetically encoded biosensor microscopy. However, the dwell time (open duration) reported in Fig 3B is indicative of the average pore lifetime. This is on the 1-minute time scale, making it more comparable to that observed via fluorescence biosensing.

Fluorescent calcium biosensing is limited by binding affinity of the biosensor. However, we have examined the reviewer's question with a new dataset recording calcium fluctuations at increased frequency (new Supplementary Fig. 4). We did discern flare events that show $t_{1/2}$ of 3.5 seconds on average; flares can peak in as short as 6 secs. However, further increasing biosensor expression and instrument gain (to resolve even faster events) may not be beneficial, as it would result in significant buffering of endogenous calcium signaling and noise amplification, respectively. We also surmise that ever more transient calcium responses are less likely to be effective upstream signals.

(3) I am surprised by the strong voltage dependency of the GSDMD channels. However, they are not very consistent. In whole cells, they exhibit striking outward rectification, with clear opening $>+10$ mV (Figure S3A). In lipid bilayers, they show dual rectification. How to explain this difference? Moreover, it would help if the authors can demonstrate that living cells with functional GSDMD channels have a very depolarized membrane potential relevant to the very depolarized gating voltage.

The rectification orientation may not be directly comparable between living cells and bilayer models. We surmise that other regulators (such as potassium channels), which were absent in the biophysical models, may further mediate the direction of ion flow in living cells in concert with GSDMD pores after their activation.

While the membrane is certainly depolarized as a consequence of these large pores opening, it does not necessarily mean that outside of the patch-clamp context, membrane voltage is able to "drive" these pores to effect downstream biochemistry such as pore closure. Instead, we have shown that calcium influx may drive such behavior. Thus, we feel that voltage dependence of the observed pore dynamics is not within the scope of the present work to establish that these pores are dynamic in the first place. However, we agree with the reviewer that it is an interesting observation that calls for future work and have modified the Discussion appropriately.

(4) Importantly, this voltage dependency introduces another complicating factor into the membrane phosphoinositide dependency. Do PIP2, PIP3, and DAG change the voltage dependency of GSDMD channels? This should be tested experimentally.

As the reviewer suggested, we have now added the ramp IV curves for PIP2 and PIP3-containing model bilayers in new Supplementary Fig 5 (previously Supplementary Fig. 4). The presence of these phosphoinositides do not change the rectification behavior of these pores.

(5) The authors proposed an interesting model that, unlike classic unitary channels, the large GSDMD pores may form ellipsoidal, "eye"-shaped states that are dynamically controlled by phosphoinositide compositions in the membrane. I like this model, but it also leads to the paradox when using the term

"single channel". We often define "single channels" based on the observations of unitary currents. Here they are not very unitary, and in some striking cases (Figure 3E), defies such standard in every possible way. In addition, DAG applications not only strongly reduce the current amplitude but also protein insertion, suggesting that the putative "ellipsoidal" pore might not only be collapsed but also tentatively removed from the membrane. Therefore, the authors should make efforts in clarifying this term.

We are grateful for the reviewer's keen reading and agree that the term was not ideal. For the purpose of the manuscript, "single channel" was meant to convey the concept of "single entity". We have now addressed this by referring to them as "single pores", both in this response as well as the main text.

A key implication of our results is that GSDMD pores are not unitary ion channels. We believe that the pore dynamics is a combination of two main factors: monomers may exit and re-join thereby altering the maximum pore size. And since membrane composition controls pore dynamics, we also surmise that the ellipsoidal model is not two-state but analog: pores may shut before opening fully into a circle. Therefore, unlike the uniform current characteristic of unitary single ion channels, we could observe a distribution of peak currents in these pores as shown in Fig 3D. In light of these discussions, we feel that a direct comparison between the two systems should be carefully considered. We have added this section to the Discussion.

With regards to DAG-containing models, "(pores being) tentatively removed from the membrane" should have been detectible in the electrophysiology experiments. However, the only signals in these bilayer traces were the initial "insertion" disturbance soon after protein addition into the system. Thereafter, the traces appear "flat" as presented in Fig 3E, for tens of minutes. Thus, we believe that while pore removal from the membrane is possible, it was not directly observed in our dataset.

(6) Is it possible to test the effect of PI3K activators and PLC activators, in addition to the inhibitors?

We have now examined the effects of published PI3K/PLC activators as the reviewer suggested. Using calcium biosensor imaging, we indeed observed a decrease in spontaneous flares upon PLC activator m-3m3FBS, which is consistent with the effect of PI3K inhibition. However, this is short-lived; cells displayed a sharp saturation in intracellular calcium after the quiescent first phase. The cognate negative control o-3m3FBS showed local calcium fluctuation metrics reminiscent of the PLC inhibitor. The PI3K activator 740 Y-P showed a slight dampening of GSDMD-induced calcium flares. However, in downstream cytokine response, neither m-3m3FBS nor 740 Y-P significantly altered IL-1 β release in LPS stimulated BMDM.

These new data suggest that the GSDMD feedback we described is likely driven by specific, local calcium-sensitive enzyme complement, which could not be accurately simulated by activators. m-3m3FBS is a direct but non-isoform specific activator of both calcium sensitive and insensitive PLCs (PMID: 12695532). The distinct calcium response profile observed recalls IP3R activation and release from intracellular calcium stores (PMID: 15302681). Thus, this activator likely created a signaling context independent from that we described. Similarly, 740 Y-P is a short peptide derived from PDGF receptor (PMID: 10328886). Even though an appropriate concentration was employed

(50 µg/mL), it still relies on a robust PDGF-responsive architecture, which is unrelated to the findings of the present study.

Our new data further emphasized the delicate balance and specificity of the circuit we described. They are now included in Supplementary Fig. 8.

(7) There are lots of serious issues in reference citation:

- Reference 6 and 7 are the two original publications reporting GSDMD cleavage by caspase-1 and -11 and the induction of pyroptosis, but the latter one was not cited together reference 6 in the Introduction section.
- PMID: 27281216 is the most comprehensive original publication that describes the pore-forming activity of the gasdermin family including GSDMD cleaved by caspase-1/11, but this reference was omitted (while other much less elegant papers were all cited).
- Although the key mutations (affecting phosphoinositide binding) were deduced from the information reported in PMID: 27281216; when introducing the anti-tumor activity of gasdermin pore formation and pyroptosis, PMID: 32188939 and 32299851, the two most important publications were also omitted in place of citing reference 16.
- When introducing structural analyses of gasdermins (references 30 and 31), the authors also missed two more original and elegant publications (PMID: 27281216 and PMID: 32109412) that report the auto-inhibited gasdermin structure and complex structures between GSDMD and caspase-1 (as well as caspase-11 and -4).
- As for the concept of pyroptosis, PMID: 27932073 is the first perspective article that redefines pyroptosis as gasdermin-mediated programmed necrotic cell death and as usual this reference was missing in the relevant place in this article.
- What does “may even requires apoptotic reinforcement to ensure cell death (reference 19) mean? I do not think reference 19 really has the intended meaning. Even though, PMID: 28459430 is the first reference elegantly showing that caspase-3, previously known as apoptotic caspase, can cleavage GSDME to drive the cells into pyroptosis, but unfortunately this reference was also omitted in the manuscript.

We regret the unintended oversight. We are indebted to the reviewer for identifying worthwhile improvements and have incorporated most of the suggested references where appropriate.

(8) Does the observed dynamics of pore opening/closing also apply to other gasdermins as gasdermin-B, C and E have also been shown to execute pyroptosis under important biological contexts? The reviewer is also curious whether cardiolipin, the more preferred lipid target of the gasdermin family, shows the similar property as phosphoinositide. They authors should make the effort to test this as the experiments are quite straightforward.

This contribution serves as the first demonstration of dynamics in any of the gasdermins. While the characterization of other gasdermin isoforms is beyond the scope of the present work, their dynamics is an exciting direct consequence of our report. Thus, we have updated the Discussion to promote this future work.

We are grateful for the reminder that previous work (PMID: 27281216) showed gasdermins can bind cardiolipin in vitro. As requested, we have now examined the GSDMD dynamics conferred by a straightforward replacement of phosphoinositide with cardiolipin (CL). In general, we found that CL-containing bilayer membranes do support GSDMD pore formation. The dynamics of GSDMD pores in CL containing bilayers are largely comparable to that in PIP2-containing bilayers.

As an integral part of oxidative phosphorylation, we feel that CL is not as versatile a signal mediator as phosphoinositides. And while mitochondrial poration by gasdermin has been reported (PMID: 30976076, 32164878), the primary localization of CL in inner mitochondrial membrane makes it less accessible in living cells. Preference for these curved membranes also manifest in stability of CL-containing membranes. Model bilayers containing higher than 5% of CL showed limited lifetime that prohibited pore reconstitution and electrophysiology. These data are now included as new Supplementary Fig. 6.

Minor concerns:

(1) In addition to the current amplitudes in pA, the authors should also report the conductance of putative single channels.

In response, we believe conductance (as defined by constant single ion channel output) is not ideal for understanding this system since the peak currents are not constant.

(2) In Methods, the authors described no Ca²⁺ in the bath solution for whole-cell recordings. Please check if this is true.

We apologize for the mistake. The saline bath solution contains 2.0 mM CaCl₂. It has been corrected.

(3) The text needs to be more polished. There are some clear typos and errors. For example, in p8, the sentence "the enzymes PI3K and PLC promote the relative accumulation of diacylglycerol (DAG) ..." needs to swap the position of DAG and PIP3.

We thank the reviewer for pointing out this issue. We have updated the text appropriately and paid more attention to typographical errors.

REVIEWER COMMENTS

Reviewer #1 (Remarks to the Author):

In the revised manuscript entitled “Gasdermin D Pores Are Dynamically Regulated by Local Phosphoinositide Circuitry” by Ana Beatriz Santa Cruz Garcia et al, the authors address the complex mechanisms by which the Gasdermin D (GSDMD) pores interact with target membranes, create conducting pathways, and control the transmembrane transport of physiologically relevant ions and large molecules.

As I stated in my first review, I consider the scientific content of this work highly relevant for the field. I certainly appreciate the time and effort the author put into answering my questions and providing detailed explanations in response to my comments. Addition of some of the discussion points together with experimental details to this revised version improves readability, broaden the audience, and provide interested scientists with sufficient details for replicating/expanding the described experimental work.

I have only a few minor suggestions for the authors. The claim “we detected protein insertions into the bilayers characterized via gating micro-currents” (row 150-151) is too strong. The experimental data suggest that some sort of (dielectric?) noise precedes the macroscopic ion currents. However, there is no clear-cut evidence that this noise originates in protein insertion. An alternative source might be protein binding to the membrane (no insertion), conformational changes of the protein bound to or inserted into the membrane, changes in dipolar moment or polarizability, and many others. The authors may state that “the minute current fluctuations observed after protein addition suggest membrane-protein interactions that precede pore formation”, or something similar. In the same line, I would strongly advise avoiding the term “gating currents”. This term is coined to describe the current resulted from the movement of the voltage-domain sensor of voltage-gated channels, and this may not be at all the case of Gasdermin D pores.

Reviewer #2 (Remarks to the Author):

The authors largely failed to address the first three points that I raised previously. Particularly, for the first point, the authors do not understand that extracellular LPS stimulation alone does not lead to

GSDMD activation (a firmly established knowledge in the inflammasome field). So, I am almost certain that what they are measuring in Fig. S1 and S2A have little to do in reporting physiological activation of GSDMD pores. Unfortunately, this is the foundation for the subsequent studies.

The failure of addressing the second and third points also suggests that there are potential technical flaws there that also jeopardize the conclusion of this study.

Response to referees

We wish to thank the reviewers for their helpful comments. Reviewer comments are italicized, and our response follows.

Reviewer #1 (Remarks to the Author):

In the revised manuscript entitled “Gasdermin D Pores Are Dynamically Regulated by Local Phosphoinositide Circuitry” by Ana Beatriz Santa Cruz Garcia et al, the authors address the complex mechanisms by which the Gasdermin D (GSDMD) pores interact with target membranes, create conducting pathways, and control the transmembrane transport of physiologically relevant ions and large molecules.

As I stated in my first review, I consider the scientific content of this work highly relevant for the field. I certainly appreciate the time and effort the author put into answering my questions and providing detailed explanations in response to my comments. Addition of some of the discussion points together with experimental details to this revised version improves readability, broaden the audience, and provide interested scientists with sufficient details for replicating/expanding the described experimental work. I have only a few minor suggestions for the authors. The claim “we detected protein insertions into the bilayers characterized via gating micro-currents” (row 150-151) is too strong. The experimental data suggest that some sort of (dielectric?) noise precedes the macroscopic ion currents. However, there is no clear-cut evidence that this noise originates in protein insertion. An alternative source might be protein binding to the membrane (no insertion), conformational changes of the protein bound to or inserted into the membrane, changes in dipolar moment or polarizability, and many others. The authors may state that “the minute current fluctuations observed after protein addition suggest membrane-protein interactions that precede pore formation”, or something similar. In the same line, I would strongly advise avoiding the term “gating currents”. This term is coined to describe the current resulted from the movement of the voltage-domain sensor of voltage-gated channels, and this may not be at all the case of Gasdermin D pores.

We thank the reviewer for these important reminders and have updated the manuscript as suggested. Specifically, the terms “protein insertion” and “gating current” are now removed and replace with “protein-membrane interaction” and “micro-current”, respectively where appropriate.

Reviewer #2 (Remarks to the Author):

The authors largely failed to address the first three points that I raised previously. Particularly, for the first point, the authors do not understand that extracellular LPS stimulation alone does not lead to GSDMD activation (a firmly established knowledge in the inflammasome field). So, I am almost certain that what they are measuring in Fig. S1 and S2A have little to do in reporting physiological activation of GSDMD pores. Unfortunately, this is the foundation for the subsequent studies.

With respect, we appreciate the reviewer's point that extracellular LPS stimulation per se may not lead to GSDMD activation and that LPS breaching is required for NLRP3 inflammasome activation.

However, we disagree that our work in any way contradicts existing paradigm. It is important to note that LPS does not remain extracellular indefinitely, and extracellularly administered LPS was directly visualized in the cytoplasm (PMID: 28990935). LPS is transported into the cell through endocytosis (via CD14, PMID: 26546281; and HMGB1, PMID: 30314759); CD14 mediated transport can in fact occur minutes after LPS addition. Lysosomal degradation induces breaching of LPS into the cytosol (PMID: 30332623), which in turn activates GSDMD (PMID: 33472215).

Furthermore, the detection of GSDMD activation is technique dependent. Our fluorescent biosensing displayed a high sensitivity for active GSDMD: just two dozen activated monomers will cause pore formation and local calcium influx in a single living cell. Such minute activation is well-below the detection limit of techniques such as immuno-blotting of N-terminal GSDMD fragment.

Therefore, the foundation of our work rests on the simultaneous use of biosensing and the novel optogenetic GSDMD to directly and precisely interrogate GSDMD activation, including at very low levels. As we showed in the present study, this synergy led to the observation of pore dynamics and consistently unified results from multiple perspectives in ways previously impossible. The value of this advanced approach has now been made explicit in the revision.

The failure of addressing the second and third points also suggests that there are potential technical flaws there that also jeopardize the conclusion of this study.

To recapitulate, the reviewer noted in the second point that the ON/OFF rates of individual calcium flares were tens of seconds, yet electrophysiological recordings seemed much more rapid.

With respect, the reviewer formed a misimpression by focusing on extremely fast events. However, analyses over many electrophysiology traces showed that pores have no such preference and are frequently slower (Fig. 3C). In the previous response, we pointed out that faster events have concomitantly lower calcium flux and may not be detected by endogenous enzymes as signals. Lastly, we had also emphasized that comparison across datasets showed an *agreement* between electrophysiology and fluorescent biosensing, and both showed an averaged dwell time (open duration) of tens of seconds (Fig. 3B and Fig. 5C, respectively).

In response to the third point, we respectfully disagree. The whole-cell patch clamp studies are fully consistent with and reinforce the studies using lipid bilayer models.

The control experiments shown in Supplementary Fig. 3A ("Fig. S3A") made clear that there was little current unless caspase-cleaved GSDMD proteins were added. However, after pores formed and a ramp protocol is applied, other channels on the live cell membrane could still contribute to

total current because these were living *whole-cell* experiments. Thus, we want to emphasize that Supplementary Fig. 3A simply indicated that we *could* form GSDMD pores inside live endothelial cell membrane and should not be over-interpreted. We previously stated the caveat in the legend. We have now further clarified the figure legend to avoid confusion.

To examine GSDMD specific properties, we utilized the reductionist bilayer model, where only the GSDMD pore was present. The pore showed dual rectification whether in PE/PC bilayers or in negatively-charged PIP2- and PIP3-containing bilayers (Supplementary Fig. 5). This is expected for a large, 21 nm diameter pore with little obstruction/restriction. We stated this result in lines 163-164 of the previous revision (lines 161-163 in the current revision).

Reviewers' comments:

Reviewer #1 (Remarks to the Author):

In this second revision, the authors properly addressed my concerns with regards to the use of terms customarily used to indicate more specific instances in membrane biophysics. The manuscript was adjusted accordingly, therefore potential confusions were properly eliminated.

With regards to the concerns raised by other reviewers:

A major point raised during the review process was the inability of extracellular LPS stimulation alone to lead to GSDMD activation, which is well-established knowledge in the inflammasome field. In my opinion, the authors present a good argument for potential indirect activation through a cascade of events leading to LPS breaching into cytosol. This answer may be expanded and included in the main text of the manuscript (the discussion section would be a good place). In the same line, the answers to the other two questions may be also better detailed in the main text. Demonstrating awareness of potential problems will not only improve the readability but also prompt scientists to further investigate those and provide useful insights into GSDMD activation.

Reviewer #3 (Remarks to the Author):

Comments to the authors:

In this manuscript, the authors present an interesting study aiming to describe how Gasdermin D Pores Are Dynamically Regulated by Local Phosphoinositide Circuitry. They employed optogenetic tools, live cell fluorescence biosensing, and electrophysiology et al. methods to support their conclusions. In addition to what was studied before, which was the mechanism of inflammasome-activated gasdermin D (GSDMD) causing pyroptosis by forming membrane pores and preferential release of mature interleukin-1 (PMID: 27383986; PMID: 29695864; PMID: 32943500; PMID: 33883744), a liposome leakage assay was also developed for monitoring gasdermin activity effectively (PMID: 31455540). These current findings are interesting and novel since they reveal that oligomeric GSDMD pore dynamics mediate an intricate, balanced calcium signaling to drive the opening or closing of pores and can therefore self-regulate pyroptosis. The manuscript is conceptually strong with a very mechanistic approach. There are, however, several aspects that need to be addressed in order to substantiate the statement presented in the paper.

Major Issues:

Previous studies have shown that in addition to GSDMD pore-forming, there are many other pore-forming proteins such as bacterial pore-forming toxins and immune pore-forming proteins, including the lymphocyte-killing cytotoxic granule perforin and mixed-lineage kinase domains, such as pseudokinase (MLKL) (a pore-forming protein that causes necroptosis). Perforin can form non-ion-selective β barrel-shaped pores, which are similar in size and structure to GSDMD (PMID: 31492708). When the plasma membrane is damaged by mechanical disruption or formation of large non-ion-selective pores, the usually different ion concentrations between the cytosol and the extracellular fluids will quickly balance; not only by releasing K^+ and activating the NLRP3 inflammasome, but also by Ca^{2+} and Na^+ flowing in. All cells have the capacity to trigger a rapid mechanism to repair plasma membrane damage, which is termed "cellular wound-healing response". This repair process is initiated when intracellular Ca^{2+} levels rise above $\sim 100 \mu M$ (PMID: 11331898). Studies further showed that GSDMD-NT binds to liposomes containing PS or PIPs and disrupts them in the Ca^{2+} free buffer, suggesting that GSDMD-NT oligomerization, unlike perforin oligomerization, is Ca^{2+} independent. However, the authors employed optogenetic tools, live cell fluorescence biosensing, and electrophysiology to investigate if gasdermin pores display phosphoinositide-dependent dynamics without the precision controls which could prove that Ca^{2+} flare is indeed due to GSDMD pore-forming instead of other pores, such as perforin pore forming. Without the precise controls, some major concerns are as follows:

- 1). Is it just weak auxiliary data, rather than real direct evidence that the Ca^{2+} flare is coupled with oligomeric GSDMD pore dynamics under LPS stimulation?
- 2). Is it possible that GSDMD pore coupled with the Ca^{2+} channels causes an artifact of Ca^{2+} flare?
- 3). The processes of GSDMD pore-forming and its opening and closing after the pore formation are a bit confusing. How does one distinguish them from the current data? Does DAG interfere with GSDMD pore-forming?
- 4). By using a liposome leakage assay to monitor GSDM activities in vitro, three potential lipid-binding sites at the mGSDMD NTD : 3-4A (R138, K146, R152, R154), beta 1-2 loop (R43, K44, F50, W51, K52, R54), and 1-3A (K7, K10, and K14) have been identified previously (PMID: 31097341). The authors only selected the mutants from the beta 1-2 loop region, and showed a representative calcium response curve for mutants: m1 (overall inhibited calcium response); m3 (reduced flares but retained calcium saturation); and m5 (retained flares but reduced calcium saturation). Why do the mutants of m1, m3 and m5 display different calcium responses? What about the mutants in other regions (3-4A and 1-3A)?

Minor:

1). In line 62, the title “Optogenetic GSDMD forms pores and recapitulates the phenotype of activated macrophages” only described macrophages. However, the contents of this section include fibroblasts and endothelial cells. This title needs to be changed;

2). In line 206, the authors should mention if the residues “I) R42/K43 and II) K51/R53/K55” are from mice or humans. Also, the residues should be based on the protein sequences from the NCBI database, not from the PDB database, and need to be labeled correctly.

Reviewer #1 (Remarks to the Author):

In this second revision, the authors properly addressed my concerns with regards to the use of terms customarily used to indicate more specific instances in membrane biophysics. The manuscript was adjusted accordingly, therefore potential confusions were properly eliminated.

With regards to the concerns raised by other reviewers:

A major point raised during the review process was the inability of extracellular LPS stimulation alone to lead to GSDMD activation, which is well-established knowledge in the inflammasome field. In my opinion, the authors present a good argument for potential indirect activation through a cascade of events leading to LPS breaching into cytosol. This answer may be expanded and included in the main text of the manuscript (the discussion section would be a good place). In the same line, the answers to the other two questions may be also better detailed in the main text. Demonstrating awareness of potential problems will not only improve the readability but also prompt scientists to further investigate those and provide useful insights into GSDMD activation.

We thank the reviewer for these complementary remarks. The orthogonality of our platforms mean that this issue has little impact on the science and logic presented. Thus, we have decided to omit the LPS supplementary figure. We agree with the Reviewer that additional discussion will help readers gain valuable awareness, as well as highlight the precision of our optogenetic approach, which allows for direct study of GSDMD activation, in contrast to LPS stimulation, which involves multiple steps and initiates multiple pathways. Therefore, as suggested, we have incorporated our responses into the Discussion.

Reviewer #3 (Remarks to the Author):

Comments to the authors:

In this manuscript, the authors present an interesting study aiming to describe how Gasdermin D Pores Are Dynamically Regulated by Local Phosphoinositide Circuitry. They employed optogenetic tools, live cell fluorescence biosensing, and electrophysiology et al. methods to support their conclusions.

In addition to what was studied before, which was the mechanism of inflammasome-activated gasdermin D (GSDMD) causing pyroptosis by forming membrane pores and preferential release of mature interleukin-1 (PMID: 27383986; PMID: 29695864; PMID: 32943500; PMID: 33883744), a liposome leakage assay was also developed for monitoring gasdermin activity effectively (PMID: 31455540).

These current findings are interesting and novel since they reveal that oligomeric GSDMD pore dynamics mediate an intricate, balanced calcium signaling to drive the opening or closing of pores and can therefore self-regulate pyroptosis. The manuscript is conceptually strong with a very mechanistic approach. There are, however, several aspects that need to be addressed in order to substantiate the statement presented in the paper.

We thank the reviewer for these positive and supportive comments.

Major Issues:

Previous studies have shown that in addition to GSDMD pore-forming, there are many other pore-forming proteins such as bacterial pore-forming toxins and immune pore-forming proteins, including the lymphocyte-killing cytotoxic granule perforin and mixed-lineage kinase domains, such as pseudokinase (MLKL) (a pore-forming protein that causes necroptosis). Perforin can form non-ion-selective β barrel-shaped pores, which are similar in size and structure to GSDMD (PMID: 31492708). When the plasma membrane is damaged by mechanical disruption or formation of large non-ion-selective pores, the usually different ion concentrations between the cytosol and the extracellular fluids will quickly balance; not only by releasing K^+ and activating the NLRP3 inflammasome, but also by Ca^{2+} and Na^+ flowing in. All cells have the capacity to trigger a rapid mechanism to repair plasma membrane damage, which is termed "cellular wound-healing response". This repair process is initiated when intracellular Ca^{2+} levels rise above $\sim 100 \mu M$ (PMID: 11331898). Studies further showed that GSDMD-NT binds to liposomes containing PS or PIPs and disrupts them in the Ca^{2+} free buffer, suggesting that GSDMD-NT oligomerization, unlike perforin oligomerization, is Ca^{2+} independent.

However, the authors employed optogenetic tools, live cell fluorescence biosensing, and electrophysiology to investigate if gasdermin pores display phosphoinositide-dependent dynamics without the precision controls which could prove that Ca^{2+} flare is indeed due to GSDMD pore-forming instead of other pores, such as perforin pore forming. Without the precise controls, some major concerns are as follows:

We appreciate the reviewer's comparison with cellular wound-healing response, MLKL, perforin, and other membrane events, which as the reviewer points out, are potential parallel activations that might have made it difficult to interpret the results. We have now clarified that this is the central benefit of our direct optogenetic activation, which altogether bypasses these potential complicating factors raised by the reviewer. By directly expressing and controlling GSDMD, our results does not involve other pores as they are not photoactivatable. We now clearly state these points in the Results section.

We performed mutagenesis on lipid-binding residues in GSDMD-N. While these residues are buried in the bilayer and thus have little contact with other proteins, they (as well as those suggested by the reviewer in a later comment) nonetheless directly altered the calcium response (Fig. 4). Furthermore, we have now added **new Supplementary Fig. 4** to show that following optogenetic activation, inhibition from its cognate C-terminal domain was sufficient to significantly dampen the observed calcium response. These data illustrate that GSDMD need not solicit other partners to control calcium dynamics and is the single major contributor in these observations.

Lastly, we emphasize that in vitro recombinant GSDMD pores, which we have examined on a variety of lipid bilayer compositions, cannot engage cellular wound healing nor any other pore complex. As we have detailed in the revised manuscript and previous responses, data from such a reductionist electrophysiology method remain in full agreement with live cell biosensing that GSDMD pores are dynamic and display phosphoinositide-dependent behavior.

1). Is it just weak auxiliary data, rather than real direct evidence that the Ca²⁺ flare is coupled with oligomeric GSDMD pore dynamics under LPS stimulation?

We thank the reviewer for this comment and agree that as presented, the LPS stimulation data was not substantial. As such, we have removed the supplementary figure in question.

2). Is it possible that GSDMD pore coupled with the Ca²⁺ channels causes an artifact of Ca²⁺ flare?

With respect, we believe the parsimonious explanation to the calcium observation is that a ~21 nm diameter puncture in the membrane that can release large cytokines, can also permit calcium ion influx without invoking the activity of other channels. This is supported by the GSDMD N-terminal mutagenesis data (Fig. 4) and the **new Supplementary Fig. 4**. The former shows that mutations on GSDMD itself were sufficient to alter calcium response; the latter shows that other partners are not necessary as the cognate autoinhibitory domain remains effective after opto-activation. In addition, our calcium imaging showed stochastic and spatiotemporally localized flares that does not resemble the whole-cell oscillations known from many inflammatory calcium channel.

We have now added new data to further address this comment. We directly visualized the transient fluxes of a large fluorogenic molecule that occurs concurrently, and at the same location, together with calcium flares in live cells (**new Supplementary Fig. 3**). This observation excludes ion channels, which cannot conduct large molecules, and static large pores, which cannot produce transients, from contributing. Our conclusion that GSDMD pores are dynamic can now be supported via calcium and other molecular flux. We believe the sum of the prior evidence and new data definitively show that the observed calcium dynamics are directly attributable to GSDMD dynamics.

3). The processes of GSDMD pore-forming and its opening and closing after the pore formation are a bit confusing. How does one distinguish them from the current data? Does DAG interfere with GSDMD pore-forming?

Our data indicate DAG content predisposes the closed state of the GSDMD pore. We captured clear membrane-protein interaction even on DAG-containing bilayers (revised Supplementary Fig. 8), suggesting that DAG likely does not interfere with pore-formation. However, as we mentioned in a previous response to Reviewer 1, we cannot be certain whether the inserted proteins formed functional pores. In living cells, however, our model suggests that a DAG content shift induces the close state in *already inserted* pores.

4). By using a liposome leakage assay to monitor GSDM activities in vitro, three potential lipid-binding sites at the mGSDMD NTD : 3-4A (R138, K146, R152, R154), beta 1-2 loop (R43, K44, F50, W51, K52, R54), and 1-3A (K7, K10, and K14) have been identified previously (PMID: 31097341). The authors only selected the mutants from the beta 1-2 loop region, and showed a representative calcium response curve for mutants: m1 (overall inhibited calcium response); m3 (reduced flares but retained calcium saturation); and m5 (retained flares but reduced calcium saturation). Why do the mutants of m1, m3 and m5 display different calcium responses? What about the mutants in other regions (3-4A and 1-3A)?

In response, we surmise that m1/m3/m5 mutants displayed different calcium response from each other as the precise manner in which they interfered with phospholipid binding is subtly different. The differences between these mutants in living cells could be observed both in the representative traces (Fig. 4D) as well as across many cells (Figs. 4B/C). We believe this highlights the sensitivity of the GSDMD pore dynamics is a cooperative function that amplifies monomer-level differences.

We thank the reviewer for the important comment on other potential lipid binding sites. We focused on select sites to convey the concept that altering lipid binding can directly change GSDMD pore dynamics; but we did not rule out possible contributions of other regions.

Our work is based on human GSDMD (hGSDMD); the corresponding potential binding residues in the regions α 1-3A and α 3-4A of hGSDMD is R7/R10/R11 and R137, R151/R153, respectively. We have now vastly expanded our exploration of the structure-dynamic response of hGSDMD by adding new mutant data to illustrate the effects of electrostatic charge alteration on these residues. These lipid binding site mutants showed significantly reduced pore functions compared to wild-type GSDMD pores, consistent with previous liposomal studies, but which our optogenetic approach now resolves in great detail. The mutant in the highly conserved α 1-3A region carrying R7E/10E/11E (termed "AE") showed significantly dampened total calcium response but retained smaller calcium flares. A single R137E mutation in the α 3-4A region ("BE1") could significantly slow/encumber calcium dynamics; in contrast, another α 3-4A mutant carrying R151E/R153E ("BE2,") was efficient at suppressing overall calcium response.

These new results and prior data from the β 1-2 loop region (Fig. 4) together now compose evidence from at least 3 distinct regions in GSDMD to support our conclusion that the lipid binding residues in GSDMD are highly evolved for specific pore dynamic behaviors. These data and are included as **new Supplementary Figure 9**.

Minor:

1). In line 62, the title “Optogenetic GSDMD forms pores and recapitulates the phenotype of activated macrophages” only described macrophages. However, the contents of this section include fibroblasts and endothelial cells. This title needs to be changed;

We thank the reviewer for this note and have updated the subheading title.

2). In line 206, the authors should mention if the residues “I) R42/K43 and II) K51/R53/K55” are from mice or humans. Also, the residues should be based on the protein sequences from the NCBI database, not from the PDB database, and need to be labeled correctly.

We thank the reviewer for this reminder. The optogenetic GSDMD we reported was based on human GSDMD; we had indicated this in line 206 of the previous revision as “hGSDMD” and have now emphasized this more clearly in the description of the optogenetic design. With respect, our residue numbering was not derived from the PDB database, and we have confirmed these values to be correct as presented. As the Reviewer suggested, we have now referenced the NCBI database link in Supplementary Information to clarify this protein sequence numbering scheme.

REVIEWERS' COMMENTS

Reviewer #3 (Remarks to the Author):

In the revised version of the manuscript “Gasdermin D Pores Are Dynamically Regulated by Local Phosphoinositide Circuitry”, the authors satisfactorily addressed the concerns I previously voiced.

The data regarding live cell biosensing and that GSDMD pores are dynamic and display phosphoinositide-dependent behavior is clear. Accordingly, the observation that calcium dynamics are directly attributable to GSDMD dynamics appears convincing.

However, there are still some points about the function assay that are unclear:

- 1). The authors made a series of mutants and found that calcium flares were affected. Is it possible to substitute the endogenous GSDMD with these mutants and observe the same effect? Do calcium flares change with the release of inflammatory factors after substitution of the endogenous GSDMD?
- 2). Given that calcium dynamics is the key to GSDMD dynamics, can the concentration of extracellular calcium be manipulated? It is clear that we should pay attention to comparing the presence or absence of calcium, but should we also observe the opening and closing characteristics of the pores through a series of analysis and comparison of different calcium concentrations?

This data should at least be mentioned and discussed since it would significantly increase the confidence on the proposed model.

Reviewer #3 (Remarks to the Author):

In the revised version of the manuscript “Gasdermin D Pores Are Dynamically Regulated by Local Phosphoinositide Circuitry”, the authors satisfactorily addressed the concerns I previously voiced.

The data regarding live cell biosensing and that GSDMD pores are dynamic and display phosphoinositide-dependent behavior is clear. Accordingly, the observation that calcium dynamics are directly attributable to GSDMD dynamics appears convincing.

We thank the Reviewer for the supportive comments.

However, there are still some points about the function assay that are unclear:

1). The authors made a series of mutants and found that calcium flares were affected. Is it possible to substitute the endogenous GSDMD with these mutants and observe the same effect? Do calcium flares change with the release of inflammatory factors after substitution of the endogenous GSDMD?

We thank the reviewer for the interesting perspective.

Our data showed a strong correspondence between calcium dynamics and cytokine release. We agree with the Reviewer and expect cells expressing caspase-sensitive GSDMD carrying these mutations will show blunted release of inflammatory cytokines.

IL-1 β /IL-18, the major pro-inflammatory cytokines secreted via GSDMD, activate the myddosomal NF κ B pathways. We are not aware of clear reports that upstream IL-1 β /IL-18 can alter PI3K/PLC activities. However, if released factors subsequently modify the activities of these phosphoinositide enzymes, then we could indeed expect changes in GSDMD pore and hence calcium dynamics.

2). Given that calcium dynamics is the key to GSDMD dynamics, can the concentration of extracellular calcium be manipulated? It is clear that we should pay attention to comparing the presence or absence of calcium, but should we also observe the opening and closing characteristics of the pores through a series of analysis and comparison of different calcium concentrations?

This data should at least be mentioned and discussed since it would significantly increase the confidence on the proposed model.

We appreciate the Reviewer’s concept that external calcium may control GSDMD dynamics. We had utilized the absence of extracellular calcium to show the source of observed calcium dynamics (Fig. 1D). However, the calcium-phosphoinositide circuit we described resides inside the cell and thus responds to local intracellular, not extracellular, calcium.

Intracellular calcium is homeostatically controlled by calcium stores/buffers in ER and mitochondria. If varying extracellular calcium concentrations interacts with these regulations to affect intracellular

calcium balance, it will modify the calcium-phosphoinositide dynamics we have presented and in turn GSDMD pore dynamics. Thus, one may observe some effects, albeit indirectly.

We have added the above into the Discussion as suggested.